# Dynamic recruitment of the curvature-sensitive protein ArhGAP44 to nanoscale membrane deformations limits exploratory filopodia initiation in neurons

Milos Galic[1]*[†], Feng-Chiao Tsai[1][‡], Sean R Collins[1], Maja Matis[2][§], Samuel Bandara[1][¶], Tobias Meyer[1]*

[1]Department of Chemical and Systems Biology, Stanford University, Stanford, United States; [2]Department of Pathology, Stanford University, Stanford, United States

**Abstract** In the vertebrate central nervous system, exploratory filopodia transiently form on dendritic branches to sample the neuronal environment and initiate new trans-neuronal contacts. While much is known about the molecules that control filopodia extension and subsequent maturation into functional synapses, the mechanisms that regulate initiation of these dynamic, actin-rich structures have remained elusive. Here, we find that filopodia initiation is suppressed by recruitment of ArhGAP44 to actin-patches that seed filopodia. Recruitment is mediated by binding of a membrane curvature-sensing ArhGAP44 N-BAR domain to plasma membrane sections that were deformed inward by acto-myosin mediated contractile forces. A GAP domain in ArhGAP44 triggers local Rac-GTP hydrolysis, thus reducing actin polymerization required for filopodia formation. Additionally, ArhGAP44 expression increases during neuronal development, concurrent with a decrease in the rate of filopodia formation. Together, our data reveals a local auto-regulatory mechanism that limits initiation of filopodia via protein recruitment to nanoscale membrane deformations.

*For correspondence: galic@uni-muenster.de (MG); tobias1@stanford.edu (TM)

Present address: [†]Institute of Medical Physics and Biophysics, University of Muenster, Muenster, Germany; [‡]Institute of Molecular Medicine, National Taiwan University, Taipei, Taiwan; [§]Institute for Neuro and Developmental Biology, University of Muenster, Muenster, Germany; [¶]Department of Systems Biology, Harvard University, Boston, United States

**Competing interests:** The authors declare that no competing interests exist.

## Introduction

During the development of the central nervous system, neuronal progenitor cells proliferate, migrate, and finally differentiate into functional units to form a multi-cellular neuronal network (**Ayala et al., 2007**). In culture, differentiation of individual neurons occurs in a stereotypic pattern starting with the formation of an axon, followed by the creation of an elaborate dendritic tree and culminating with the initiation and maturation of trans-neuronal synaptic contacts (**Dotti et al., 1988**). The formation of synaptic connections is often facilitated by dynamic exploratory filopodia that extend out of thicker dendritic branches to sample the environment and thereby increase the probability that selective pre-to-postsynaptic connections are established (**Ziv and Smith, 1996**; **Marrs et al., 2001**). Exploratory filopodia are dynamic finger-like membrane structures containing actin cables formed out of actin patches along the dendritic shaft (**Lau et al., 1999**; **Matus, 2000**). Extension of filopodia is driven by local activation of formins, Ena/VASP proteins, small GTPases, and likely other steps (**Krugmann et al., 2001**; **Lebrand et al., 2004**). Intriguingly, the frequency of filopodia formation dramatically drops once high synapse density is established (**Ziv and Smith, 1996**), suggesting that the initiation of these structures is controlled by opposing negative regulators. However, the identity of these inhibitors has remained elusive.

Here, we provide evidence that ArhGAP44 limits the initiation of exploratory dendritic filopodia. Consistent with previous reports, we observe that formation of actin patches precedes filopodia extension. Notably, we find that within actin patches Myosin II-mediated pulling on plasma

**eLife digest** Our brains contain a vast network of many billions of cells that communicate with, and are connected to, each other. Each brain cell, or neuron, can form connections with as many as 10,000 other neurons—and signals pass from one neuron to the next at sites known as synapses.

A neuron's surface has numerous finger-like protrusions known as filopodia that are important for sensing the environment around the cells. Filopodia are highly changeable and constantly extend and retract as the filaments that support them—which are made up of a protein called actin—grow and shrink back. Neurons use their filopodia to explore and seek out other neurons in the brain, and when they make contact with the right neuron, it leads to the formation of a synapse. However, how filopodial extensions begin to grow—and what stops a neuron from forming too many filopodia—is not fully understood.

Galic et al. now show that a protein called ArhGAP44 limits the formation of new filopodia in neurons. The ArhGAP44 protein is recruited to patches of the surface membrane that have a lot of actin and that curve inwards. ArhGAP44 then locally inhibits other proteins that are normally required to extend the actin filaments and drive the growth of filopodia out from the surface of the cell.

Galic et al. also show that more ArhGAP44 is produced with age—levels are low in embryos and high in adults—and this increase in the amount of protein correlates with a decrease in the number of filopodia formed. When Galic et al. engineered rat neurons to produce more of the ArhGAP44 protein, fewer filopodia formed on the surface of the neurons. Decreasing the amount of this protein had the opposite effect. Moreover, ArhGAP44 was shown to mainly stop new filopodia from forming and had little effect on existing filopodia. Together, these findings suggest that ArhGAP44 may help neurons transition from a dynamic exploratory mode to a mature, more static, state; this is a characteristic of the development of the nervous system.

membrane (PM)-associated actin cables induces highly curved membrane sections that trigger ArhGAP44 recruitment. The resulting enrichment of ArhGAP44 then reduces local actin polymerization due to the Rac GAP activity of ArhGAP44, preventing the formation of filopodia. ArhGAP44 expression increases as the neuronal network is established and the frequency of exploratory filopodia formation is diminished, suggesting that ArhGAP44 may facilitate the transition of neurons from a dynamic exploratory mode to a mature more static state, a hallmark of nervous system development.

## Results

### ArhGAP44 is predominantly expressed in the brain and increases with age

Formation of filopodia depends on proteins that regulate polymerization of actin filaments (*Krugmann et al., 2001*; *Lebrand et al., 2004*). To identify new regulators of exploratory dendritic filopodia formation, we performed a literature search and identified 286 genes that were previously associated either directly or indirectly with actin reorganization. We then clustered these genes according to expression pattern using published microarray data and found 89 of the 286 genes to be expressed predominantly in neuronal tissues (*Figure 1—figure supplement 1A* and 'Materials and methods' and *Supplementary file 1*). As we were interested in regulators of actin dynamics selective for the brain, we ranked these 89 genes for high expression in the brain compared to the spinal cord and tested the validity of the ranking using a set of control genes expressed only in one of the respective tissues (*Figure 1—figure supplement 1B* and *Table 1* and 'Materials and methods'). Among the five actin regulators with the highest brain vs spinal cord ratio, we found ArhGAP44 (also known as Rich2 or Nadrin2), a membrane-curvature-sensing GTPase Activating Protein (GAP) selective for the small Rho GTPases Rac1 and Cdc42 (*Richnau and Aspenstrom, 2001*). In previous studies, ArhGAP44 has been associated with the maintenance of apical microvilli in polarized epithelial cells as well as postsynaptic maturation and vesicle release (*Rollason et al., 2009*; *Nahm et al., 2010*; *Raynaud et al., 2013*, *2014*). Considering the role of Rho GTPases during filopodia formation (*Krugmann et al., 2001*), we decided to further investigate the role of ArhGAP44 in developing neurons.

**Table 1.** Reference genes expressed predominantly in the adult brain or in the spinal cord

| Gene | Adult brain | Spinal cord | Ratio | Reference |
|---|---|---|---|---|
| PMP22* | 720.55 | 2516.35 | 0.286347 | (*Snipes et al., 1992*) |
| MPZ* | 5.5 | 181.5 | 0.030303 | (*Su et al., 1993*) |
| BSN† | 56.7 | 3.45 | 16.43478 | (*tom Dieck et al., 1998*) |
| GRIA2† | 355.1 | 11.55 | 30.74459 | (*Martin et al., 1993*) |

*enriched in the spinal cord.
†enriched in the adult brain.

To validate the microarray expression pattern of ArhGAP44, we first isolated various organs and brain regions and probed protein levels with an antibody directed against ArhGAP44. Consistent with previous work (*Richnau and Aspenstrom, 2001*), western blot analysis showed expression of ArhGAP44 in the brain while being below detection level in all other tested organs (*Figure 1A* and *Figure 1—figure supplement 2A*). Within the brain, immunoblotting directed against ArhGAP44 showed increased protein levels in the frontal cortex and olfactory bulb (*Figure 1B*). The same expression pattern was found in sagittal brain sections stained against ArhGAP44 (*Figure 1—figure supplement 2B*).

We then explored the expression of ArhGAP44 in the brain during development and found ArhGAP44 among the genes with the highest adult-to-fetal ratio (*Figure 1—figure supplement 3*, and *Table 2* and 'Materials and methods'). To validate the observed increase in ArhGAP44 expression with age, we measured protein levels in rat primary hippocampal neurons by western blot, using samples isolated during neurite extension (3 days in vitro, i.e., DIV3), the peak of exploratory filopodia formation (DIV10), and after initial synaptic contacts were formed (DIV17). Consistent with the microarray data, ArhGAP44 protein levels increased with time (*Figure 1C*).

As N-BAR domains present in ArhGAP44 and other proteins can bind to inward-curved plasma membranes (*Galic et al., 2012*), we used high-resolution Field Emission Scanning Electron Microscopy to examine potential changes to local neuronal membrane curvature during maturation. As expected, we observed an increase in overall neuronal complexity with time (*Figure 1—figure supplement 4*). High magnification micrographs further showed convoluted membrane sections (i.e., convoluted nodes) along the dendrite at DIV10 that may reflect such local regions of high curvature (*Figure 1D* and *Figure 1—figure supplement 5*).

## ArhGAP44 negatively regulates filopodia density
To determine the function of ArhGAP44 in neurons, we transfected primary rat hippocampal neurons. Compared to control cells, overexpression of ArhGAP44 caused a significant reduction in the density of dendritic filopodia at DIV12 (*Figure 1E*, dark blue). Prolonged expression increased the fraction of cells forming varicosities, likely due to increased RhoGAP activity associated with elevated ArhGAP44 levels (*Figure 1—figure supplement 6*). To decrease the GAP activity of ArhGAP44, we substituted a conserved arginine in the catalytic cleft with a methionine, which was shown to reduce but not eliminate the enzymatic activity of GAP proteins (*Muller et al., 1997*; *Graham et al., 1999*). Consistent with partial activity, expression of the ArhGAP44(R291M) mutant reduced but did not abolish GTP hydrolysis of the small GTPase Rac1 (*Figure 1F*), showed a milder loss in filopodia density (*Figure 1E*, light blue), and delayed the onset of varicosity formation (*Figure 1—figure supplement 6*).

To further validate the observed effect on filopodia density, neurons were transfected with siRNA directed against ArhGAP44. Markedly, knockdown of ArhGAP44 augmented the density of dendritic filopodia (*Figure 1G*, yellow). Control experiments using a second siRNA pool directed against a different region of ArhGAP44 mRNA confirmed the phenotype, thus showing that siRNA knockdown and overexpression of ArhGAP44 have opposing effects. Both siRNAs were effective since protein levels of ArhGAP44 were significantly reduced in cells co-transfected with either one of the siRNA pools directed against ArhGAP44 (*Figure 1H*).

## ArhGAP44 negatively regulates de novo filopodia formation
As dendritic filopodia frequently extend, reorient, and collapse (*Ziv and Smith, 1996*), filopodia density reflects the product of de novo formation frequency and stabilization rate. To determine which of

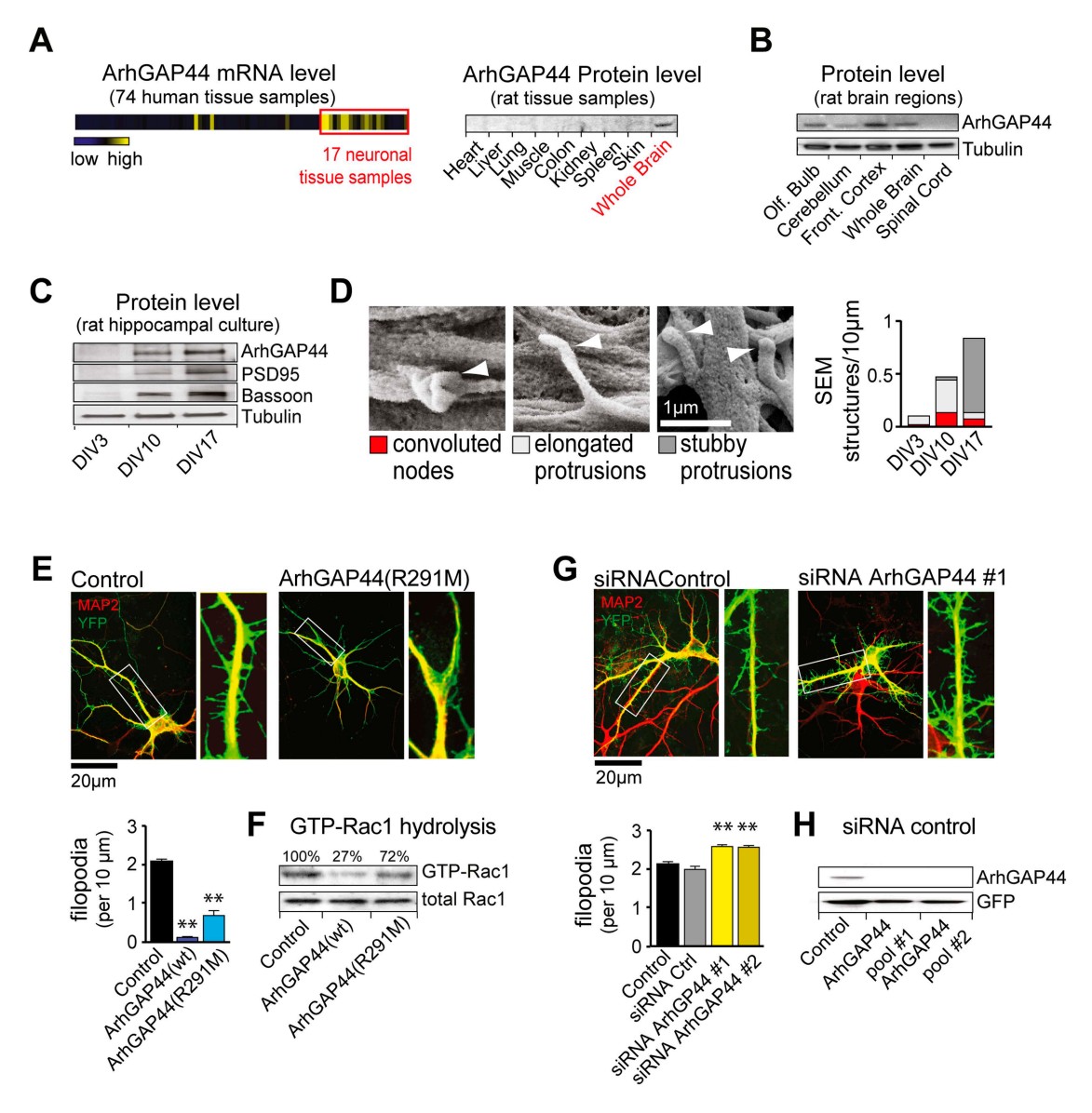

**Figure 1**. The brain-enriched ArhGAP44 regulates exploratory dendritic filopodia formation. (**A**) Microarray data for ArhGAP44 across 74 tissue samples show predominant expression in neuronal tissues (red box). Immunoblot to the right shows ArhGAP44 expressed in the brain while being below detection level in other tissues. Note that individual tissue samples are likely composed of a variety of different cell types. (**B**) Total extracts of individual brain regions probed with an antibody directed against ArhGAP44 (top) and tubulin (bottom). (**C**) Expression of ArhGAP44 in cultured hippocampal neurons. Total extracts of individual neuronal samples were isolated at DIV3, DIV10, and DIV17 and probed with an antibody directed against ArhGAP44 (top), the synaptic proteins PSD95 and Bassoon (middle lanes), and tubulin (bottom). (**D**) Scanning electron micrographs of cultured neurons. Dendritic surface structures are classified based on morphology as convoluted nodes (red), elongated protrusion without contact (light gray), or stubby protrusions that contact adjacent neurites (dark gray). Analysis is shown for DIV3 (n = 10 neurons, 2 independent experiments), DIV10 (n = 10 neurons, 2 independent experiments), and DIV17 (n = 9 neurons, 2 independent experiments). (**E**) Overexpression of ArhGAP44 decreases filopodia density. Representative examples of neurons (green) stained with anti-MAP2 antibody (red) are shown. Analysis of filopodia density upon overexpression of control (black; n = 67 neurons, 3 independent experiments), ArhGAP44(wt) (dark blue; n = 73 neurons, 3 independent experiments), and mutant ArhGAP44(R291M) (light blue; n = 53 neurons, 3 independent experiments) is shown below. (**F**) Rac-GAP activity of individual ArhGAP44 mutants. Note that ArhGAP44(R291M) shows higher GTP-Rac1 hydrolysis than control. (**G**) Knockdown of ArhGAP44 increases filopodia density. Analysis of filopodia density upon expression of control (black; n = 67 neurons, 3 independent experiments), control siRNA (gray; n = 67 neurons, 3 independent experiments), and knockdown of ArhGAP44 (siRNA #1, light yellow; n = 83 neurons; siRNA #2, dark yellow; n = 85 neurons; both 3 independent experiments) is shown below. (**H**) Control western blot analysis testing the effectiveness of individual siRNA pools. Scale bars (**D**), 1 μm; (**E** and **F**), 20 μm.

*Figure 1. Continued on next page*

*Figure 1. Continued*

The following figure supplements are available for figure 1:

**Figure supplement 1**. Cluster analysis of putative actin-regulating genes.

**Figure supplement 2**. Ponceau loading control of various tissues and ArhGAP44 protein expression in the brain.

**Figure supplement 3**. ArhGAP44 expression increases over time.

**Figure supplement 4**. Neuronal complexity increases over time.

**Figure supplement 5**. Electron micrographs of neurons.

**Figure supplement 6**. Overexpression phenotypes of ArhGAP44 in cultured neurons.

these parameters are controlled by ArhGAP44, we performed time-lapse imaging of dendritic filopodia dynamics (*Figure 2A* and *Figure 2—figure supplement 1*). Compared to control, neither knockdown of ArhGAP44 nor expression of ArhGAP44(R291M) showed significant changes in the density of static protrusions that persisted longer than 10 min (*Figure 2B* and *Figure 2—figure supplement 2A* and *Video 1*) In contrast, expression of ArhGAP44(R291M) reduced (*Figure 2B*, blue and *Video 2*), whereas knockdown of ArhGAP44 increased (*Figure 2B*, yellow and *Video 3* and *Video 4*) the formation of dynamic protrusions. Most of these newly formed protrusions were short-lived, while stabilization of extending protrusions or collapse of previously static protrusions was observed only infrequently (*Figure 2B* and *Figure 2—figure supplement 2B*). Together, these results argue that ArhGAP44 primarily limits filopodia formation with little effect on the stabilization of existing filopodia.

For both, overexpression and knockdown of ArhGAP44, we further observed an increase in the number of transiently formed nodes along dendritic arbors (*Figure 2C* and *Figure 2—figure supplement 3* and *Videos 2–5*). Intriguingly, we find that the majority (83% ± 7%) of de novo protrusions extended from such node-like structures (*Figure 2—figure supplement 4* and *Video 6*). We thus consider these nodal structures to represent nascent filopodia sites, where filopodia formation is either initiated or aborted. Given their spacing and their presence at the same time in culture, they likely correlate with the convoluted nodes along dendrites seen in electron microscopy studies (*Figure 1D*).

Consistent with previous reports (*Richnau and Aspenstrom, 2001*), we find that the GAP domain of ArhGAP44 inhibits GTPase activity of Rac and Cdc42 (*Figure 2—figure supplement 5A*). We thus aimed to investigate to which extent the two possible targets of ArhGAP44 contribute to filopodia formation. We find that overexpression of wild-type Rac (human Rac1) but not Cdc42 phenocopied

**Table 2.** Reference genes expressed predominantly in the adult or in the fetal brain

| Gene | Adult brain | Fetal brain | Ratio | Reference |
|---|---|---|---|---|
| RELN* | 34.85 | 93.35 | 0.373326 | (*D'Arcangelo et al., 1995*) |
| DCX* | 10.55 | 2577 | 0.004094 | (*des Portes et al., 1998*) |
| NRXN1* | 72.1 | 143.15 | 0.503667 | (*Ushkaryov et al., 1992*) |
| NLGN1* | 7.5 | 24.7 | 0.303644 | (*Ichtchenko et al., 1995*) |
| CAMK2B† | 818.15 | 321.25 | 2.54677 | (*Omkumar et al., 1996*) |
| MUNC13† | 26.3 | 9.75 | 2.697436 | (*Betz et al., 1998*) |
| MECP2† | 818.15 | 321.25 | 2.54677 | (*Amir et al., 1999*) |
| PSD95† | 362.1 | 12.7 | 28.51181 | (*Kornau et al., 1995*) |
| PACSIN1† | 90.8 | 20.1 | 4.517413 | (*Qualmann et al., 1999*) |

*enriched in the fetal brain.
†enriched in the adult brain.

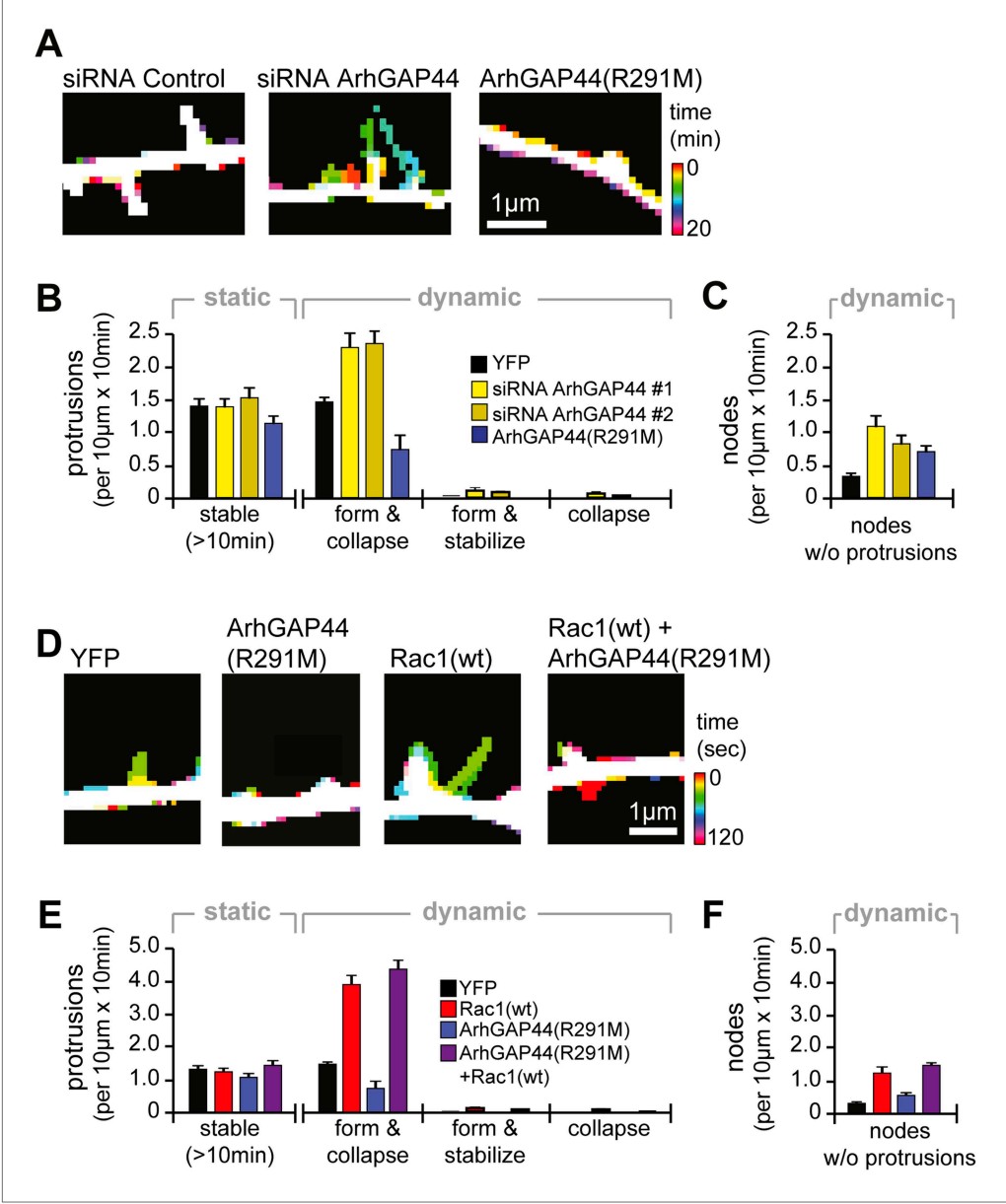

**Figure 2**. Knockdown of ArhGAP44 and overexpression of Rac both increase de novo filopodia formation.
(**A**) Color-coded overlay of image-series. Note increased protrusion dynamics upon knockdown of ArhGAP44.
(**B** and **C**) ArhGAP44 negatively regulates de novo protrusions. Analysis of protrusion dynamics in neurons
transfected with control (black; n = 85 protrusions, 10 neurons, 3 independent experiments), ArhGAP44(R291M)
(blue; n = 106 protrusions, 11 neurons, 3 independent experiments), and upon knockdown of ArhGAP44 (yellow;
siRNA #1 = 131 protrusions, 12 neurons, 3 independent experiments; siRNA #2 = 215 protrusions, 15 neurons,
3 independent experiments). Compared to controls, overexpression of ArhGAP44(R291M) reduces while knock-
down of ArhGAP44 increases formation of transient protrusion (**B**). Both increase node formation (**C**). (**D**) Color-
coded overlay of image-series. Note increased protrusion dynamics upon overexpression of Rac1. (**E** and **F**)
Co-expression of Rac1 reverses ArhGAP44-dependent reduction in protrusion dynamics. Compared to control
(black; n = 85 protrusions, 10 neurons, 3 independent experiments), overexpression of Rac1 (red, n = 123 protru-
sions, 9 neurons, 3 independent experiments) increases the formation of transient protrusion (**E**) and nodes (**F**).
For both parameters, co-expression of Rac1 with ArhGAP44(R291M) (purple, n = 126 protrusions, 12 neurons,
3 independent experiments) can compensate for the reduction observed for ArhGAP44(R291M) alone (blue,
n = 106 protrusions, 11 neurons, 3 independent experiments). Scale bars (**A** and **D**), 1 μm.

*Figure 2. Continued on next page*

*Figure 2. Continued*

The following figure supplements are available for figure 2:

**Figure supplement 1**. Filopodia analysis tool and GTPase overexpression control.

**Figure supplement 2**. Examples of prototypic dendritic protrusions.

**Figure supplement 3**. Examples of dendritic nodes.

**Figure supplement 4**. Acquisition interval of 60 s is sufficient to detect dynamic protrusions but not all nodes.

**Figure supplement 5**. The ArhGAP44 knockdown phenotype is phenocopied by the small GTPase Rac1 but not Cdc42.

protrusion density (*Figure 2—figure supplement 5B*) as well as protrusion kinetics (*Figure 2—figure supplement 5C* and *Videos 7–10*) observed upon knockdown of ArhGAP44. Similar to knockdown of ArhGAP44, Rac1 overexpression increased protrusion dynamics and node formation (*Figure 2D–F* and *Figure 2—figure supplement 5D* and *Videos 7,8*). These observations are consistent with previous work showing Rac activity associated with increased actin patch formation and filopodia dynamics in axons (*Spillane et al., 2012*) and dendrites (*Korobova and Svitkina, 2010*; *Cheadle and Biederer, 2012*). We thus considered that ArhGAP44 might act by inhibiting Rac to limit initiation of exploratory filopodia formation.

To test this hypothesis, we performed a synthetic compensation experiment, co-expressing Rac1 with ArhGAP44(R291M). Although the resulting protrusions were shorter than control filopodia, co-expression of ArhGAP44(R291M) together with Rac1 reversed the observed decrease in the frequency of protrusion formation and node formation caused by expression of ArhGAP44(R291M) (*Figure 2E,F* purple and *Video 11*).

**Video 1**. Example of a stable dendritic protrusion. Neuron was transfected with a fluorescence marker at DIV11 and imaged 24 hr later. Individual frames were taken every 60 s. Scale bar is 2 μm. Video is 720× real-time.

## ArhGAP44 localizes to patches that precede filopodia extension

To analyze the subcellular protein localization, we cultured hippocampal neurons and stained against endogenous ArhGAP44. We found ArhGAP44 to be absent from the nucleus and present in patches along dendrites (*Figure 3—figure supplements 1 and 2A*). We then expressed fluorescently tagged ArhGAP44 in cultured neurons. Like the endogenous protein, fluorescently tagged ArhGAP44 was excluded from the nucleus, distributed uniformly through the cytosol, and formed distinct ArhGAP44 patches along dendrites (*Figure 3—figure supplement 2B,C*), arguing that the fluorescent tag did not interfere with its localization.

To investigate the molecular mechanism that caused ArhGAP44 patch formation, we compared the subcellular localization of various deletion mutants of ArhGAP44 (*Figure 3A*). ArhGAP44 contains an N-BAR domain that has been reported to bind to positively (i.e., inward) curved lipid membranes (*Peter et al., 2004*). Full-length protein and the isolated N-BAR domain of ArhGAP44 both enriched in patches along dendritic arbors, while deletion of the amino-terminal amphipathic

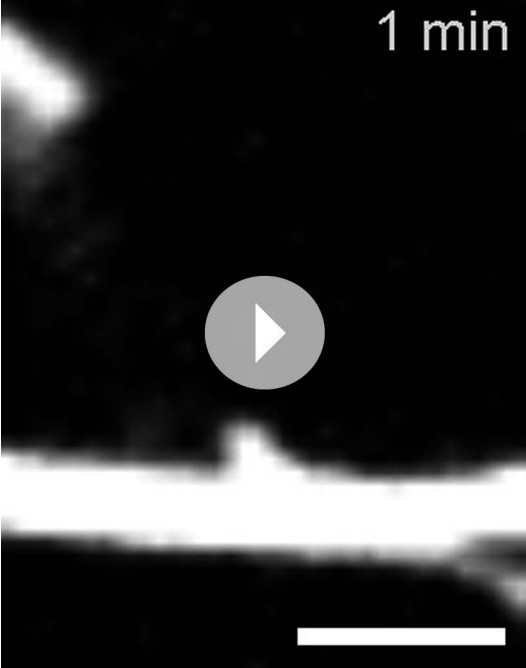
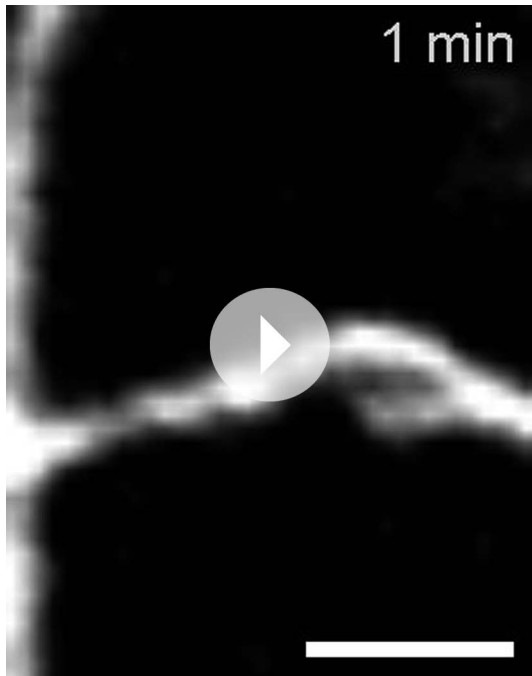

**Video 2**. ArhGAP44(R291M) transfection causes increased dendritic node and reduced protrusion formation. Neuron was transfected with ArhGAP44(R291M) at DIV11 and imaged 24 hr later. Individual frames were taken every 60 s. Scale bar is 2 μm. Video is 360× real-time.

**Video 3**. Knockdown of ArhGAP44 increases dendritic node and protrusion formation. Neuron was co-transfected with diced RNA directed against ArhGAP44 and a cytosolic reference at DIV7 and imaged at DIV12. Individual frames were taken every 60 s. Scale bar is 2 μm. Video is 360× real-time.

helix (ΔN-BAR), a critical sequence motif for binding and stabilization of curved membranes (*Peter et al., 2004*), showed no enrichment over a cytosolic reference (*Figure 3B*). Intriguingly, none of the ArhGAP44 constructs enriched in extended filopodia (*Figure 3C*). Time-lapse imaging showed that filopodia often emerged from ArhGAP44 patches (*Figure 3D*). We considered that ArhGAP44 patches might reflect the convoluted, node-like dendritic membrane sections we previously observed in electron micrographs (*Figure 1D*) and fluorescence images (*Figure 2—figure supplement 3A*). Consistently, measurement of individual membrane folds within nodes (*Figure 3—figure supplement 2D*) showed sufficient deformation to trigger curvature-dependent protein recruitment to the plasma membrane (*Bhatia et al., 2009*). Together with the previous overexpression and knockdown experiments (*Figure 2A,B*), the transient localization of ArhGAP44 to patches but not to extended filopodia argues that ArhGAP44 limits initiation rather than elongation of newly formed filopodia.

## Myosin-dependent contraction of PM-associated actin filaments induces membrane curvature and ArhGAP44 recruitment

Next, we aimed to investigate what caused the convoluted membrane surface at dendritic nodes. Consistent with previous reports (*Lau et al., 1999*; *Spillane et al., 2012*), ratio-imaging of the filamentous actin marker f-tractin (*Johnson and Schell, 2009*) to a cytosolic reference showed formation of actin patches that preceded extension of exploratory filopodia (*Figure 4—figure supplement 1A* and *Video 12*). This was the case in 89% ± 6% of all filopodia (*Figure 4—figure supplement 1B*). To directly test whether actin patches cause convoluted node-like PM subsections (*Figure 3—figure supplement 2D*), we performed correlative light and electron microscopy (*Figure 4—figure supplement 2*). Consistent with a role of actin in node-formation, we find that actin intensity in nodes was significantly higher compared to adjacent dendritic sections (*Figure 4A*). We then expressed f-tractin together with the isolated N-BAR domain of ArhGAP44 and found significant enrichment of f-tractin in ArhGAP44 patches (*Figure 4B* and *Figure 4—figure supplement 3*), arguing that ArhGAP44 and

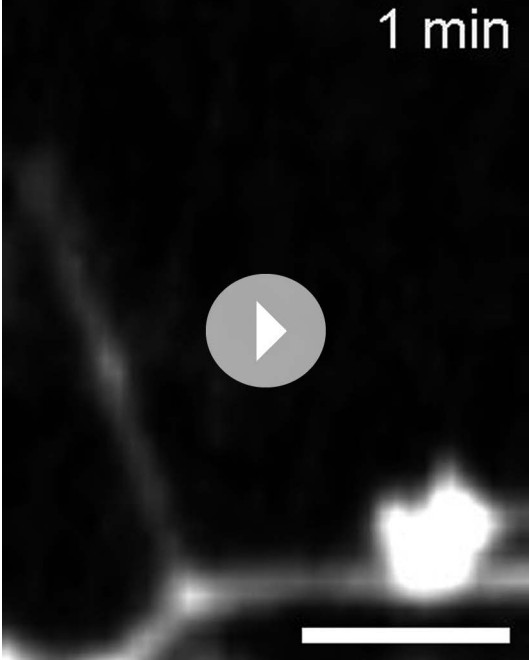

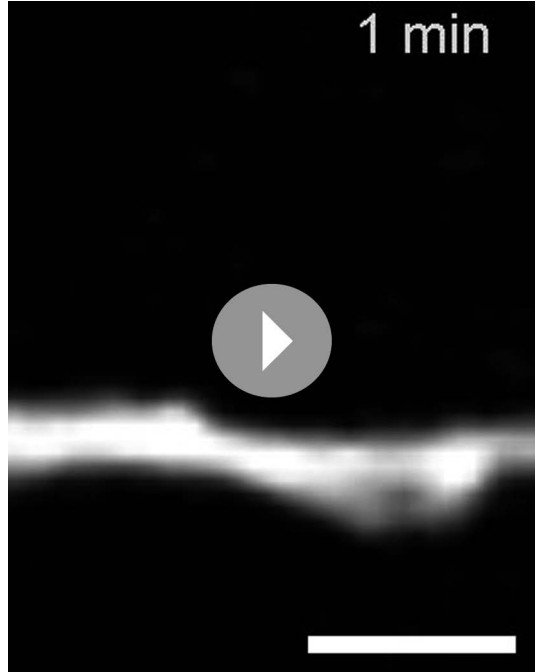

**Video 4**. Knockdown of ArhGAP44 increases dendritic node formation. Neuron was co-transfected with diced RNA directed against ArhGAP44 and a cytosolic reference at DIV7 and imaged at DIV12. Individual frames were taken every 60 s. Scale bar is 2 µm. Video is 360× real-time.

**Video 5**. Example of dendritic node formation. Neuron was transfected with a fluorescence marker at DIV11 and imaged 24hr later. Individual frames were taken every 60 s. Scale bar is 2 µm. Video is 360× real-time.

actin localize to the same structure. Consistently, the N-BAR domain of ArhGAP44 enriched in dendritic actin-patches by 80% ± 8% compared to adjacent dendritic regions (*Figure 4—figure supplement 3B*). To test for Myosin II-dependent contractile forces in ArhGAP44 patches, we co-expressed the isolated N-BAR domain of ArhGAP44 and non-muscle Myosin Heavy Chain IIB (NMHC-2B) and found a significant enrichment of NMHC-2B in ArhGAP44 patches (*Figure 4C*). Together, this suggests that myosin-dependent contraction of actin patches triggers inward membrane deformations within nodes to which ArhGAP44 is enriched. Consistently, we find increased staining for the phosphorylated form of myosin light chain (pMLC), a marker for active Myosin II (*Tan et al., 1992*), in dendritic actin patches (*Figure 4D*).

To test the hypothesis that ArhGAP44 enrichment relies on acto-myosin-dependent pulling forces to the PM, we altered either actin integrity or myosin-dependent actin contraction. Notably, both treatments led to a significant reduction of ArhGAP44 concentration in actin patches (*Figure 5A,B* and *Figure 5—figure supplement 1*), suggesting that Myosin II-dependent forces within dendritic actin patches trigger local inward deformation of the PM, mediating a recruitment of curvature-sensing protein ArhGAP44. Notably, we find ArhGAP44 to be also enriched at inward plasma membrane deformation created by retracting lamellipodia (*Figure 5—figure supplement 2* and *Video 13*), as well as in response to membrane bending by artificial nanocone structures (*Figure 5—figure supplements 3,4*; and *Videos 14,15*), arguing that inward membrane deformation is sufficient for binding of ArhGAP44 to the plasma membrane.

Finally, we investigated whether ArhGAP44 is also enriched to other actin-rich structures in neurons. We find the N-BAR domain enriched at the end of dendrites (dendritic tips; *Figure 5—figure supplement 5*), as well as in spine-shaped dendritic protrusions (*Figure 5—figure supplement 6*). Knockdown of ArhGAP44 in aged neurons facilitates re-emergence of exploratory dendritic protrusions (*Figure 5—figure supplement 7*). Thus, considering that ArhGAP44 expression increases with time, this suggests that ArhGAP44 may facilitate the transition of neurons from a dynamic exploratory mode to a mature more static state.

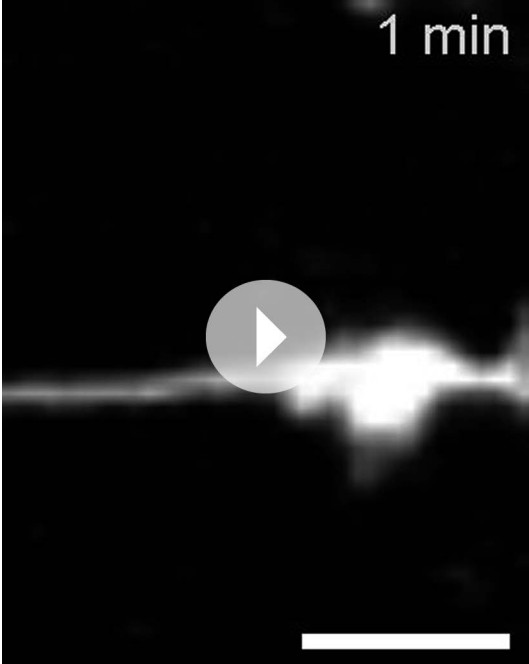
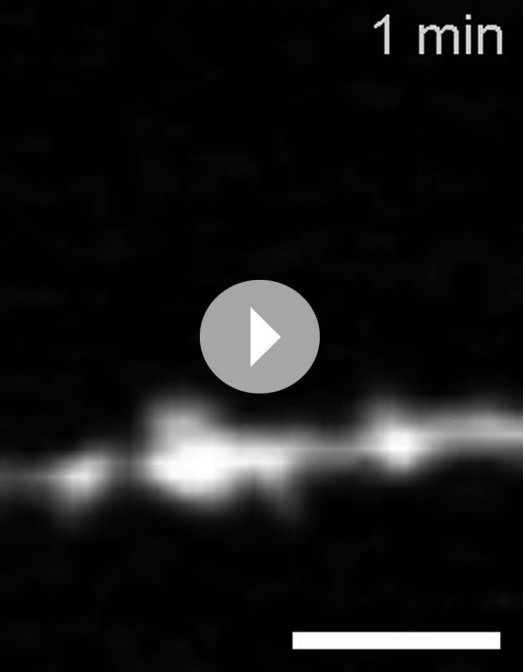

**Video 6**. Knockdown of ArhGAP44 triggers formation of protrusion from dendritic nodes. Neuron was co-transfected with RNA directed against ArhGAP44 and a cytosolic reference at DIV7 and imaged at DIV12. Individual frames were taken every 60 s. Scale bar is 2 μm. Video is 360× real-time.

**Video 7**. Rac1 overexpression causes abnormal dendritic node and protrusion formation. Neuron was transfected with Rac1(wt) at DIV11 and imaged 24 hr later. Individual frames were taken every 60 s. Scale bar is 2 μm. Video is 360× real-time.

## Discussion

Our study shows that recruitment of ArhGAP44 to actin patches, which seed exploratory filopodia along dendritic branches, is mediated by Myosin II-dependent contraction of membrane-associated actin cables. These actin patches have been shown to serve as precursors for the formation of filopodia in axons (*Lau et al., 1999*; *Spillane et al., 2012*) and dendrites (*Korobova and Svitkina, 2010*). Studies in non-neuronal cells showed that individual actin filaments within the actin cortex form bundles prior to filopodia elongation (*Svitkina et al., 2003*). Consistently, electron micrographs of neurons showed that these filopodial actin bundles are embedded in the underlying dendritic actin meshwork (*Korobova and Svitkina, 2010*). It has been proposed that once enough actin filaments are bundled to generate the force required to protrude the PM, the resulting outward membrane deformation triggers recruitment of actin bundling/Cdc42 activating proteins, which then further increase the polymerization rate within the extending filopodia (*Krugmann et al., 2001*; *Disanza et al., 2006*). Given the localization of Myosin II in actin patches (*Figure 4D*), it is reasonable to conjecture that Myosin II-dependent contraction of individual actin filaments within actin patches provides structural integrity to counter the force generated by the extending filopodia. However, considering that individual actin filaments within a bundle are oriented with the barbed end to the PM, Myosin II-dependent contraction also exerts an inward directed pull forces that curve the PM inward which triggers increased recruitment of ArhGAP44. We propose that the highly convoluted membrane topography associated with actin patches (*Figure 4A*) reflect such Myosin II-dependent contraction of membrane-associated actin cables, at which ArhGAP44 becomes enriched in a curvature-dependent manner. In support of this hypothesis, we find not only that inward membrane deformation is sufficient for ArhGAP44 and N-BAR domain recruitment (*Figure 5—figure supplements 3 and 4*), but that deletion of ArhGAP44 curvature-sensitivity (*Figure 3A,B*) or reducing action-myosin-dependent contractile forces (*Figure 5A*) both prevented ArhGAP44 enrichment.

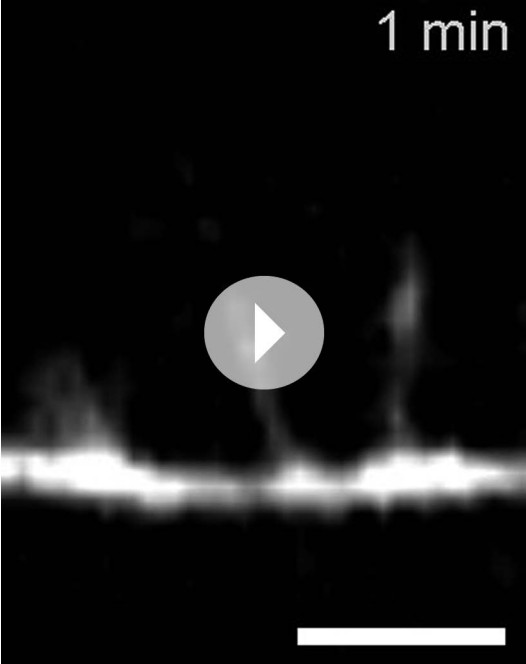
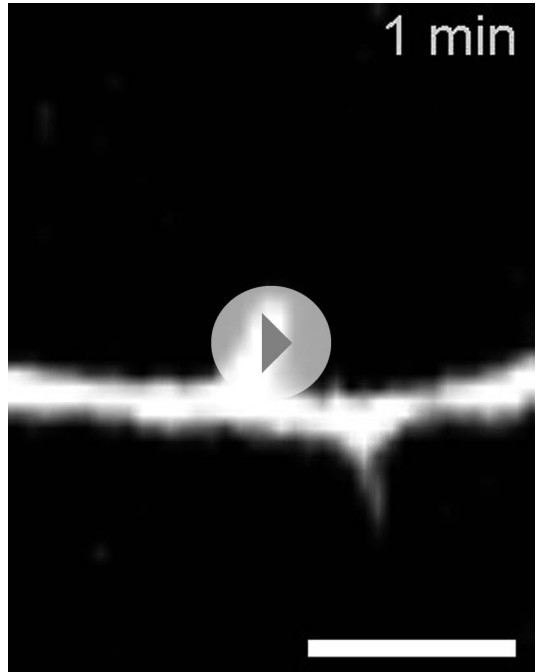

**Video 8**. Rac1 overexpression causes abnormal dendritic node and protrusion formation. Neuron was transfected with Rac1(wt) at DIV11 and imaged 24 hr later. Individual frames were taken every 60 s. Scale bar is 2 µm. Video is 360× real-time.

**Video 9**. Cdc42 overexpression causes abnormal dendritic protrusion formation. Neuron was transfected with Cdc42(wt) at DIV11 and imaged 24 hr later. Note the elongation of a preexisting dendritic protrusion. Individual frames were taken every 60 s. Scale bar is 2 µm. Video is 360× real-time.

A second major finding of our study is that recruitment of ArhGAP44 to plasma membrane deformations in nodes (i.e., acto-myosin patches) limits initiation of exploratory filopodia. We show that ArhGAP44 can hydrolize the small GTPase Rac and Cdc42 (*Figure 2—figure supplement 5*). Thus, a likely function of local ArhGAP44 at actin patches is to suppress GTPase-mediated local actin polymerization. We propose that ArhGAP44 is limiting Rac-dependent formation of actin patches, which provide the structural integrity required for filopodia to protrude outwards (*Figure 6*). This is consistent with previous reports in neurons, showing that dynamic rearrangements within actin patches relies on Rac activity (*Andersen et al., 2005*; *Spillane et al., 2012*), and that elevated Rac levels increase filopodia dynamics (*Luo et al., 1996*; *Nakayama et al., 2000*; *Zhang and Macara, 2006*; *Cheadle and Biederer, 2012*). In support of this notion, we find Rac1 localized at actin patches (*Figure 6—figure supplement 1A*) and show that artificial decrease of ArhGAP44 concentration at actin patches either by reducing overall ArhGAP44 levels by knockdown (*Figure 1F*) or by preventing ArhGAP44-recruitment to actin patches via inhibition of acto-myosin contraction (*Figure 6—figure supplement 1B*, see also [*Ryu et al., 2006*; *Hodges et al., 2011*]) both increased the number of exploratory filopodia.

Cdc42 acts as an activator of Irsp53 (*Kast et al., 2014*), promoting IRSp53-dependent enrichment and clustering of VASP and other factors to drive actin assembly in elongating filopodia (*Disanza et al., 2013*). Consistently, knockdown of Cdc42 substantially reduces filopodia formation in neurons (*Garvalov et al., 2007*). Intriguingly, overexpression of Cdc42 is not sufficient to initiate filopodia formation in neurons (*Figure 2—figure supplement 5*, see also [*Hotulainen et al., 2009*]) or in other cell lines (*Kast et al., 2014*). This has led to the hypothesis that elongation of filopodia is a combinatorial process requiring multiple factors (*Kast et al., 2014*). We propose that signal integration at actin patches controls this decision of filopodia elongation. Considering that actin-patch formation occurs before filopodia elongation, this argues for a 2-step process where Rac1-induced patch formation (and ArhGAP44-dependent regulation thereof) precedes Cdc42-induced filopodia elongation (*Figure 6*). However, since ArhGAP44 shows dual specificity for Rac1 and Cdc42, both steps will be limited by recruitment by ArhGAP44 to actin patches.

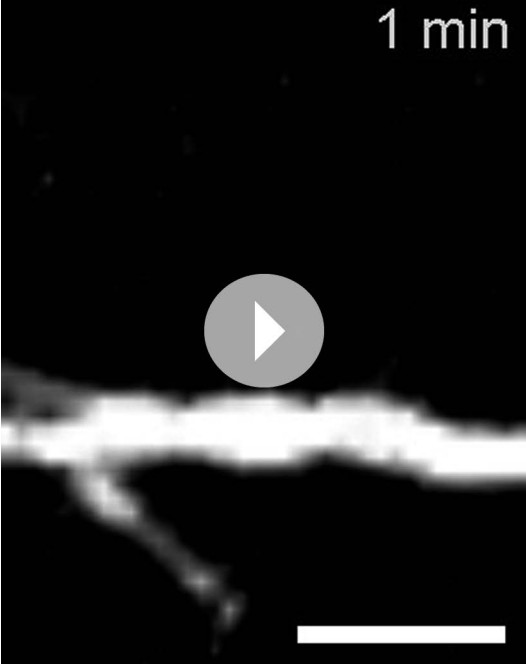

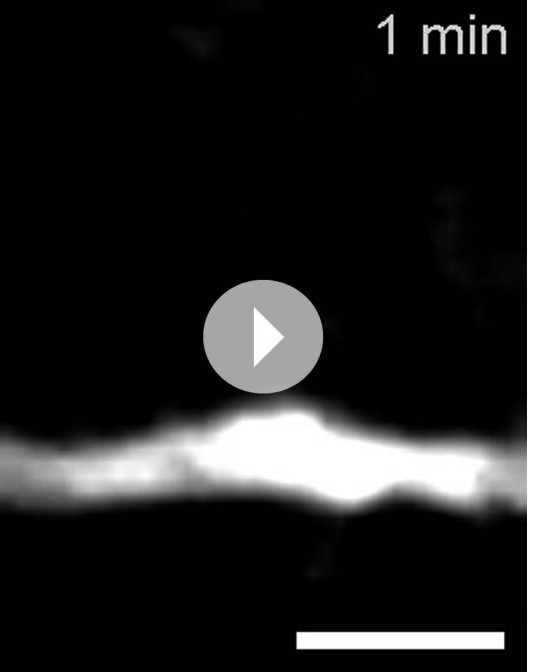

**Video 10**. Cdc42 overexpression causes abnormal dendritic protrusion formation. Neuron was transfected with Cdc42(wt) at DIV11 and imaged 24 hr later. Note the elongated filopodia emerging from the dendrite. Individual frames were taken every 60 s. Scale bar is 2 μm. Video is 360× real-time.

**Video 11**. Rac1 synthetically rescues ArhGAP44(R291M)-dependent reduction in protrusion formation. Neuron was co-transfected with Rac1(wt) and ArhGAP44(R291M) at DIV11 and imaged 24 hr later. Individual frames were taken every 60 s. Scale bar is 2 μm. Video is 360× real-time.

Taken together, we propose that ArhGAP44 mediates a localized negative feedback that becomes upregulated as neurons mature to reduce the frequency with which neurons initiate new exploratory filopodia. Considering that acto-myosin initiated PM deformation is a ubiquitous process and that a high number of curvature-sensing proteins are known to modify actin dynamics, this suggests that the local feedback mechanism for ArhGAP44 described in our study likely exemplifies a more general principle for receptor-independent signaling whereby signal transduction is initiated by the transient recruitment of regulatory proteins to actin- and force-dependent nanoscale PM indentations.

## Materials and methods

### Clustering of microarray data

A set of 286 genes relating to the actin cytoskeleton and GTPase signaling was identified with a search on the NCBI Gene database with the query 'actin AND GTPase AND human[orgn]'. The Human U133A/GNF1H Gene Atlas data set (gnf1h-gcrma unaveraged) was downloaded from the biogps.org website (*Su et al., 2004*) (only the U133A data were analyzed). The data were renormalized using the median intensity on each array. To generate a focused expression data set for actin- and GTPase-related genes, we extracted the data from all probe sets corresponding to the 286 genes described above. The data were then log-transformed, and the mean log-expression for each probe across all tissue types was subtracted to yield relative expression values. The values were then hierarchically clustered using the Cluster 3.0 software, with the Pearson correlation distance, and average linkage.

### Ranking and validating of microarray data

To test the validity of the brain vs spinal cord ranking in *Figure 1—figure supplement 1B*, we characterized genes known to be expressed exclusively in the brain or in the spinal cord (*Table 1* and [*Snipes et al., 1992*; *Martin et al., 1993*; *Su et al., 1993*; *tom Dieck et al., 1998*]). We found for the Schwann cell-specific genes MPZ and PMP22 an adult brain/spinal cord ratio smaller than 0.3 while the brain

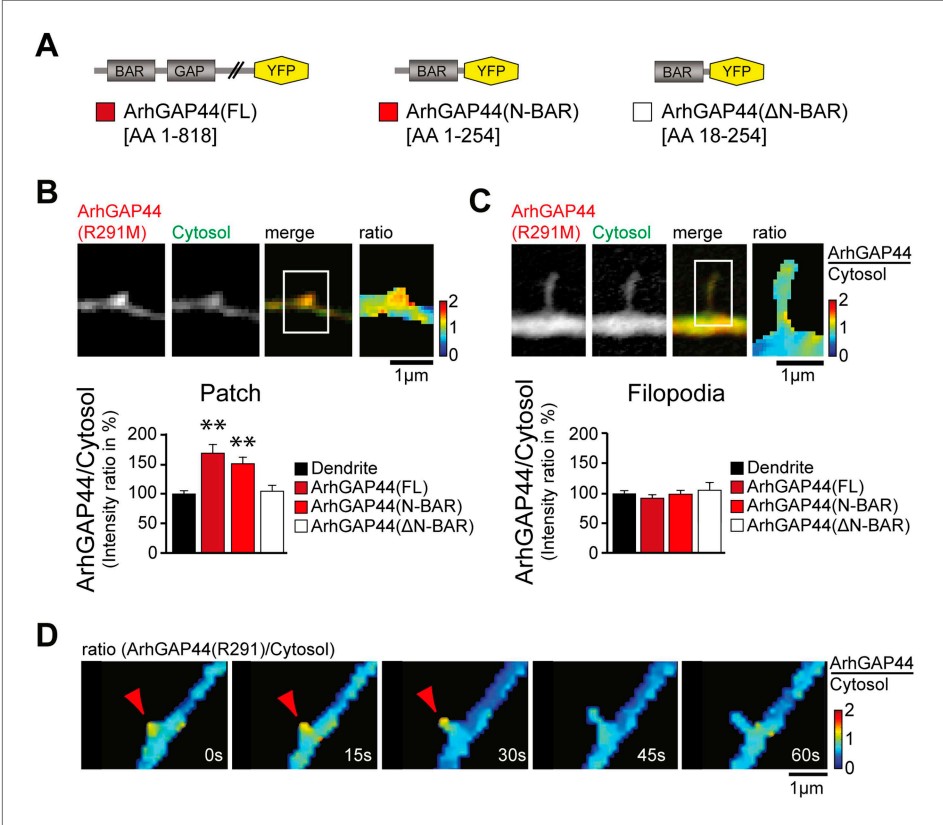

**Figure 3**. ArhGAP44 recruitment to dendritic nodes precedes filopodia extension. (**A**) ArhGAP44 protein structure and deletion mutants. ArhGAP44 is composed of an amino-terminal curvature-sensing N-BAR domain (AA 1–254), followed by the RhoGAP domain (AA 255–445) and a stretch of ~350 amino acids (AA 445–818) with no annotated domain structure. Full length (left, dark red), the isolated N-BAR domain (middle, red), and the N-BAR domain lacking the initial 18 amino acids (right, white) were tested. (**B**) Enrichment of full length (dark red; 25 patches, 12 neurons, 3 independent experiments) and the isolated N-BAR domain of ArhGAP44 (red, n = 34 patches, 11 neurons, 3 independent experiments) in dendritic patches. No enrichment is observed for the N-BAR domain lacking the first 18AA that encode an amphipathic helix critical for curvature sensing (white; 20 patches, 11 neurons, 3 independent experiments). Note, ArhGAP44(R291M) was used to reduce the compromised cell health caused by overexpression of equal levels of active wild-type protein. (**C**) No enrichment of full length (dark red; 26 filopodia, 14 neurons, 3 independent experiments), the isolated N-BAR domain of ArhGAP44 (red, n = 40 filopodia, 13 neurons, 3 independent experiments), or the N-BAR domain lacking the first 18AA (white; 22 patches, 10 neurons, 3 independent experiments) in dendritic filopodia. (**D**) Filopodia emerge from ArhGAP44-rich patches. Scale bars (**B**, **C**, **D**), 1 µm.

The following figure supplements are available for figure 3:

**Figure supplement 1**. ArhGAP44 antibody controls.

**Figure supplement 2**. ArhGAP44 localization in neurons.

enriched presynaptic vesicle fusion protein Bassoon (BSN), and the postsynaptic AMPA receptor 2 (GRIA2) both showed an adult brain/spinal cord ratio greater than 15. Of the 89 neuron-enriched genes identified in *Figure 1—figure supplement 1B* the top five hits were: the RhoGEF Kalirin that was shown to contribute to EphB receptor-dependent spine maturation (*Penzes et al., 2003*), the synaptic vesicle-associated protein Amphiphysin (*David et al., 1996*), the small GTPase K-Ras that translocates from the PM to the Golgi complex, and early/recycling endosomes in response to neuronal activity (*Fivaz and Meyer, 2005*), the microtubule tip-tracking protein EB3 that is a modulator of spine morphology (*Jaworski et al., 2009*), and ArhGAP44.

To test the validity of the ranking for genes critical for specific steps during neuronal developmental in *Figure 1—figure supplement 3A*, we characterized where genes known to be involved in various

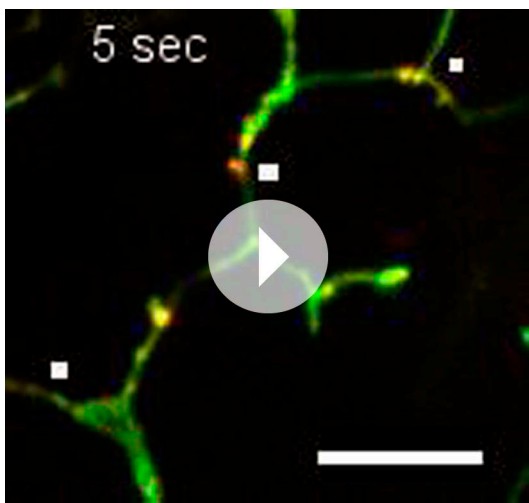

**Video 12**. Actin enriches at patches that precede exploratory filopodia initiation. Neurons were transfected with a marker for filamentous actin (f-tractin, red) and a cytosolic reference (green). Note the formation of actin patches prior to filopodia formation (white boxes). Individual frames were taken every 5 s. Video is 50× real-time.

aspects of neuronal maturation (*Table 2* and [*Ushkaryov et al., 1992*; *D'Arcangelo et al., 1995*; *Ichtchenko et al., 1995*; *Kornau et al., 1995*; *Omkumar et al., 1996*; *Betz et al., 1998*; *des Portes et al., 1998*; *Amir et al., 1999*; *Qualmann et al., 1999*]). We observed a clear separation of DCX and RELN, which are involved in neuronal migration, and NLGN1 and NRXN1, which initiate trans-synaptic contact, from MUNC13, MECP2, PSD95, and PACSIN1which all contribute to synapse function. Of the 89 neuron-enriched genes identified in *Figure 1—figure supplement 3A*, genes with the highest adult-to-fetal ratio included the microtubule tip-tracking protein EB3 (*Jaworski et al., 2009*), the MAP kinase ERK1 as well as the MAP kinase kinase MEK1 which both control dendrite development (*Crino et al., 1998*) and synaptic plasticity (*Dash et al., 1990*), the Armadillo-like protein PKP4 (*Wolf et al., 2006*), the Lowe syndrome protein OCRL (*Attree et al., 1992*), and ArhGAP44.

## Western blot analysis of rat tissue samples

Tissue samples were isolated from female Wistar rat and suspended in ice-cold lysis buffer containing 1% Tween and protease inhibitors (Roche [Indianapolis, IN], 11873580001). Each sample was homogenized and absolute protein concentration was measured, using the BCA Protein Assay Kit (Thermo Scientific [Rockford, IL], 23225), and adjusted to equal levels for each sample. Next, 6× SDS was added, and the samples were heated to 90°C for 5 min. Finally, the samples were vortexed and loaded on a gel. 20 µg total protein was loaded for each sample in *Figure 1A* and probed with an antibody directed against ArhGAP44 (Sigma-Aldrich [St. Louis, MO], HPA038814). No single protein was used as reference for comparison of ArhGAP44 expression level across organs, as the expression of conventional housekeeping proteins (e.g., tubulin or GAPDH) can vary between tissues by up to an order of magnitude (*Figure 1—figure supplement 2A* and http://biogps.org/#goto=genereport&id=37238). In *Figure 1B* 20 µg total protein was loaded and probed with an antibody directed against ArhGAP44 as well as beta-tubulin (Sigma, T8578) as a reference. For detection, secondary antibodies from Invitrogen and SuperSignal West Femto Maximum Sensitivity Substrate (Pierce [Thermo Scientific, Rockford, IL], 34095) were used.

## Westernblot analysis of cultured neurons

Neurons were harvested at DIV3, DIV10, and DIV17 in ice-cold lysis buffer containing 1% Tween and protease inhibitors (Roche, 11873580001). Absolute protein concentration was immediately measured using the BCA Protein Assay kit (Thermo 23225) and adjusted to equal levels for each time point. Relative protein levels were probed using specific antibodies directed against ArhGAP44 (Abcam [Cambridge, MA], ab93627), the postsynaptic marker PSD95 (EMD Millipore [Billerica, MA], MAB1596), the presynaptic protein Bassoon (Abcam, 76065), and the loading control beta-tubulin (Sigma, T8578). For detection, we used secondary antibodies from Invitrogen and SuperSignal West Femto Maximum Sensitivity Substrate (Pierce, 34095).

## Scanning electron micrographs of cultured neurons

Neurons were cultured on Poly-L-Lysine-coated glass coverslips and fixed using 2% Glutaraldehyde (8% stock-EM grade) and 4% p-Formaldehyde in NaCacodylate buffer pH 7.4 for 10 min. Neurons were rinsed with 0.1 M NaCacodylate buffer (pH 7.4) after primary fixation and post-fixed for 1 hr with aqueous 1% $OsO_4$, washed briefly with water and dehydrated in an ascending ethanol series (50, 70, 90, and 100% [twice] for 20 min each) before critical point drying with liquid $CO_2$ in a Tousimis 815B (Tousimis, Rockville, MD, USA). Samples were mounted on colloidal Graphite on 15-mm aluminum stubs (Ted Pella, Redding, CA, USA) and sputter-coated with 70A of Au/Pd using a Denton Desk 11

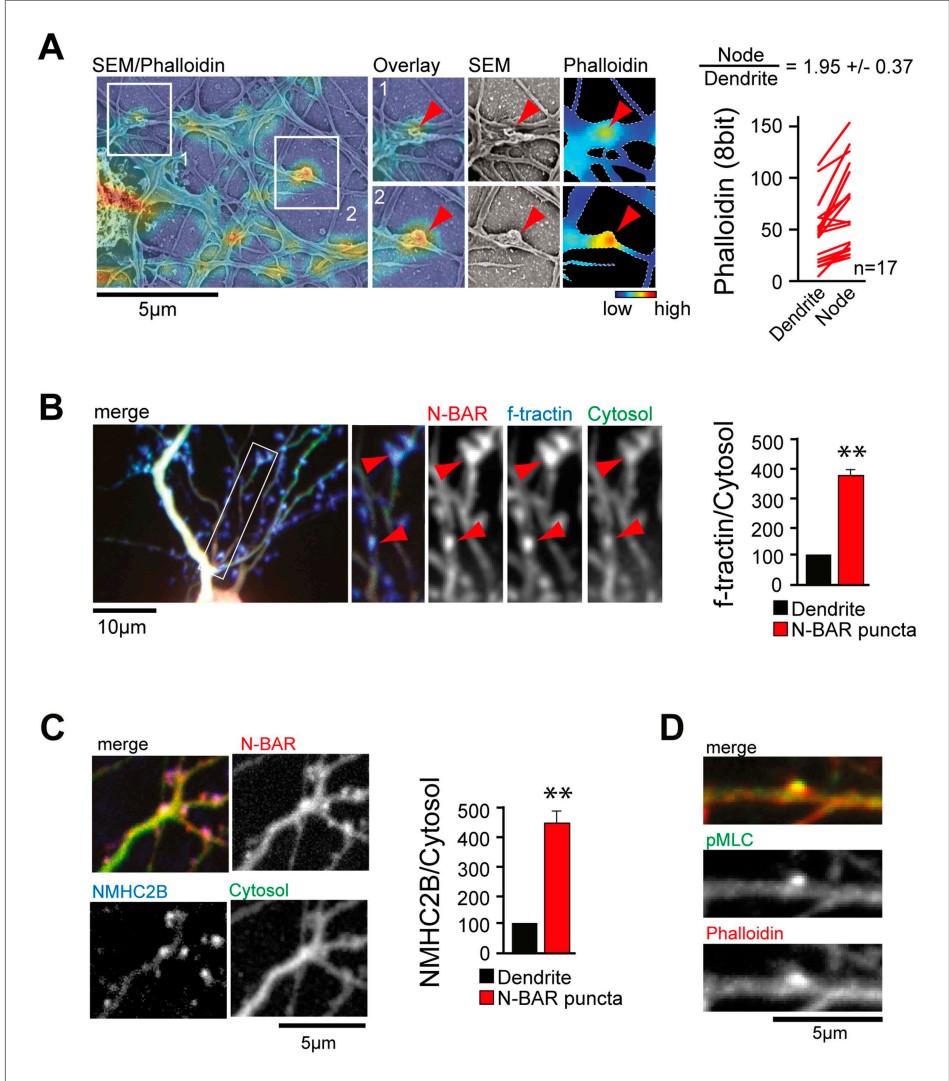

**Figure 4**. ArhGAP44 is recruited to convoluted dendritic PM sections enriched in polymerized actin and myosin. (**A**) Correlative fluorescence and scanning electron microscopy shows actin enrichment in convoluted nodes that form along dendritic arbors. Quantification of the relative fluorescent intensity of phalloidin in individual nodes compared to adjacent dendritic sections is shown to the right. (**B**) F-tractin and ArhGAP44 co-localize. Neuron was transfected with the N-BAR domain of ArhGAP44 (red), f-tractin (blue), and a cytosolic reference (green). A magnified section (white box) and quantification of relative f-tractin intensity in ArhGAP44 patches are shown next to it (34 patches form 25 neurons; 3 independent experiments). (**C**) NMHC-2B and ArhGAP44 co-localize. Neurons were transfected with the N-BAR domain of ArhGAP44 (red), NMHC-2B (blue), and a cytosolic reference (green). Quantification of relative NMHC-2B intensity in ArhGAP44 patches is shown next to it (n = 20 patches form 13 neurons; 3 independent experiments). (**D**) Phosphorylated regulatory myosin light chain (green) is enriched in dendritic actin patches (red). Scale bars (**A**, **C**, **D**), 5 μm; (**B**), 10 μm.

The following figure supplements are available for figure 4:

**Figure supplement 1**. Time-lapse of exploratory dendritic filopodia emerging from actin patches.

**Figure supplement 2**. Workflow to identify individual neurons for correlative SEM/IF microscopy.

**Figure supplement 3**. ArhGAP44 is enriched at actin patches.

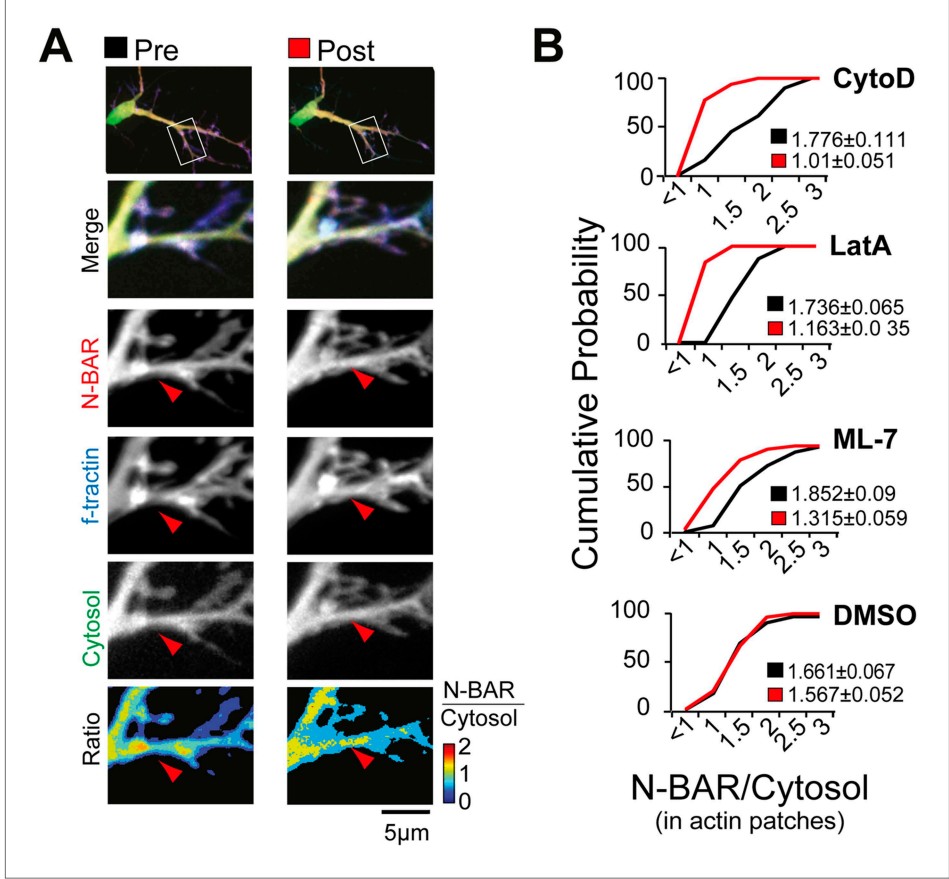

**Figure 5**. Myosin dependent recruitment of the N-BAR domain of ArhGAP44 to actin patches. (**A**) Inhibition of acto-myosin dependent forces decreases relative concentration of the N-BAR domain of ArhGAP44 in actin patches. Neurons were transfected with the N-BAR domain of ArhGAP44 (red), f-tractin (blue), and a cytosolic reference (green). Neuron is shown before (left panels) and after addition of the MLCK inhibitor ML-7 (right panels). (**B**) Cumulative distribution and average values of the relative intensity of the N-BAR domain of ArhGAP44 to a cytosolic reference in actin patches are shown before (black) and after (red) addition of CytoD (n = 49 patches; 12 cells, 3 independent experiments), LatA, (n = 27 patches; 10 cells, 3 independent experiments), ML-7 (n = 118 patches; 15 cells, 3 independent experiments), or the vehicle DMSO (n = 62 patches; 12 cells, 3 independent experiments). Scale bar, 5 μm.

The following figure supplements are available for figure 5:

**Figure supplement 1**. Inhibition of actin polymerization decreases relative concentration of the N-BAR domain of ArhGAP44 but does not completely dissolve actin patches.

**Figure supplement 2**. ArhGAP44 is recruited to collapsing artificial lamellipodia in neurons.

**Figure supplement 3**. Artificial membrane deformations recruit ArhGAP44 to the plasma membrane.

**Figure supplement 4**. N-BAR domain of ArhGAP44 enriched at nanocone-induced dendritic plasma membrane deformations in neurons.

**Figure supplement 5**. ArhGAP44 is enriched at dendritic tips.

**Figure supplement 6**. ArhGAP44 is enriched at dendritic spines.

**Figure supplement 7**. Knockdown of ArhGAP44 alters protrusion morphology and dynamics in aged neurons.

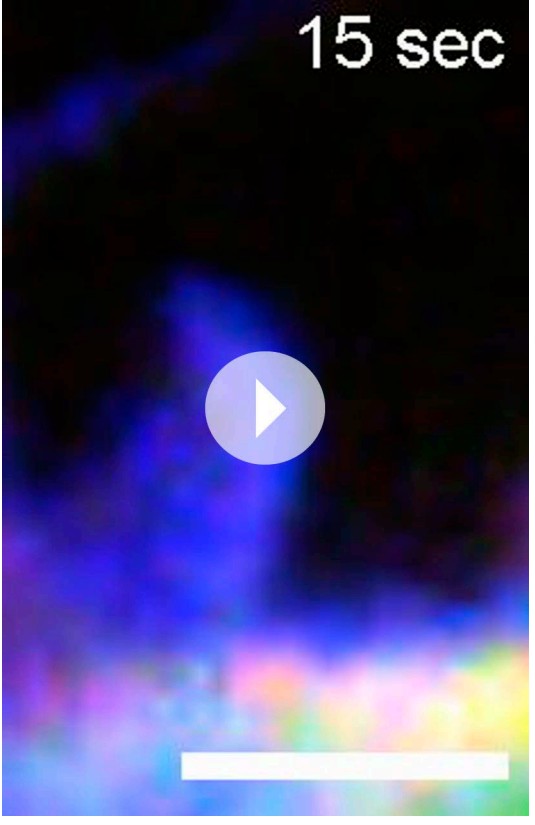

**Video 13**. Enrichment of the N-BAR domain of ArhGAP44 at retracting actin-rich structures in neurons. Neuron was transfected with Lyn-FRB, CFP-FKBP-Tiam1 (blue), the N-BAR domain of ArhGAP44 (red), and a cytosolic reference (green) at DIV11. Images show formation and retraction of artificial lamelipodia-like structures that formed along the dendritic shaft upon rapamycin-triggered recruitment of the Rac GEF Tiam1 to the plasma membrane. Individual frames were taken every 15 s. Scale bar is 2 µm. Video is 120× real-time.

Sputter Coater. Visualization of samples was performed with a Zeiss Sigma FESEM (Zeiss Microscopy LLC, Thornwood, NY) operated at 2–3 kV, working distance 4–6 mm and an in-lens SE detector under high vacuum conditions. Images were captured in TIFF format.

## Quantification of protrusion types via scanning electron micrographs

Neurons were cultured on glass slides for various periods of time (3, 10, and 17 days), fixed and prepared for SEM as described above. Using low resolution (1000× magnification), individual neurons were identified (*Figure 1—figure supplement 5A*, left panel). Starting from the soma, initial segments of the dendritic arbors were imaged at high resolution (10,000×), and individual protrusions were classified based on morphology (*Figure 1— figure supplement 5A*, right panel). Only the proximal 50–60 µm of the dendritic arbors that can clearly be associated to a particular neuron were analyzed. Examples of dendritic nodes are shown in *Figure 1—figure supplement 5B*.

## Culturing and immunostaining of primary hippocamal neurons

Rat hippocampal neurons were prepared as previously described (*Fink et al., 2003*). Neurons were transfected using Lipofectamine 2000 (Invitrogen, Carlsbad, CA) according to the manufacturer's protocol. For live imaging, neurons were plated in chambers (Lab-Tek 155383; Thermo, Rockford, IL) using NBM (Neurobasal Medium, Gibco [Life Technologies, Carlsbad, CA], 21103-049), supplemented with SM1 (StemCell Technologies [Vancouver, Canada], 05711), Pen/Strep (Gibco, 15070-063) and 20 mM HEPES (Gibco, 15630). For immunostaining, cells were fixed in PBS (Gibco, 10010-023) containing 4% Formaldehyde (Ted Pella, 18505) and 120 mM sucrose, stained, and imaged. ArhGAP44 antibody was from Sigma (1:150, HPA038814), MAP2 antibody was from Chemicon (1:1000, AB 5622; Chemicon, EMD Millipore, Billerica, MA), pMLC antibody was from Cell Signaling (1:200, 3671S; Cell Signalling Technology, Danvers, MA), and Rac1 antibody was from Cytoskeleton (1:200, ARC03; Cytoskeleton, Denver, CO).

### Fluorescence microscopy

All experiments were performed on a spinning disc confocal microscope. CFP, YFP, and mCherry excitations were obtained by a 442-nm helium cadmium laser (100 mM; Kimmon Electrics, Centennial, CO), a 514-nm argon laser (300 mW; Melles Griot, Carlsbad, CA), and a 594-nm solid-state laser (80 mW; CNI Laser, Changchun, China), respectively. Images were captured using an EMCCD camera (QuantEM 512SC [Photometrics, Tucson, AZ]), driven by Micromanager mounted on the side port of an inverted microscope (model IX-71; Olympus, Center Valley, PA).

### Analysis of filopodia density

Primary cultured rat hippocampal neurons were transfected 7 days after plating with siRNA directed against ArhGAP44 together with a fluorescent marker and fixed at DIV12. Fluorescently tagged

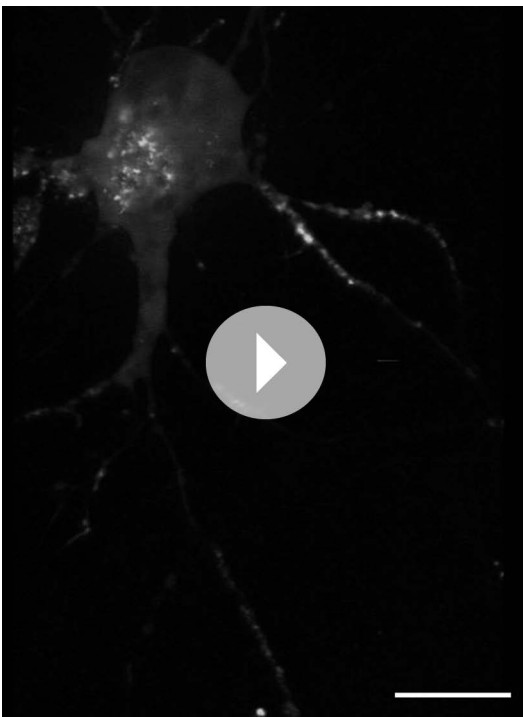

**Video 14**. Enrichment of the N-BAR domain of ArhGAP44 at basal membrane sections indented by nanocones. 3D rotation of neuron plated on nanocone-coated glass slide and transfected with the N-BAR domain of ArhGAP44. Note the punctate enrichment at the basal membrane below the soma. Scale bar is 10 μm. DOI: 10.7554/eLife.03116.046

ArhGAP44(R291M) was transfected at DIV11 and fixed at DIV12. For each condition, the sample was fixed and individual neurons were imaged. Filopodia density was measured manually, analyzing only the proximal 100 μm of each dendrite. For each condition tested, >20 cells were used.

## Rac1 activation assay

HeLa cells were grown in DMEM (high glucose) supplemented with 10% FCS and Pen/Strep until they reached 80% confluency and then transfected either with CFP-ArhGAP44(wt), CFP-ArhGAP44(R291M), or empty CFP plasmid (control) using LF2000 according to the manufacturer's protocol for 4 hr in DMEM in the absence of FCS and Pen/Strep. Cells were then serum starved for 18 hr. For all conditions, live cell fluorescence 18 hr post transfection showed transfection efficiency of >80%. Cells were stimulated with 50 ng/ml EGF for 5 min. Next, cells were scratched and protein levels were measured and adjusted for all samples to equal levels. Of each sample 900 μl were used for pulldown and 100 μl for loading control. GTP levels were probed using Rac1 Activation Assay Kit (Cell Biolabs [San Diego, CA], STA-404) according to the manufacturer's protocol. In brief, GTP-bound Rac was eluted from cell lysates using PAK PBD agarose beads and detected by western blot using α-Rac1 (Cell Biolabs, 240106) antibody. Relative intensities were compared to loading controls.

## Generation of diced siRNA pools

The protocol used to synthesize siRNA has been previously reported (*Liou et al., 2005*). Specific primers for ArhGAP44 were automatically designed and used to amplify from a cDNA library an approximate 600-bp PCR fragment of the 3′ region of the coding sequence. A second amplification was performed with a set of nested primers bearing a T7 promoter sequence on their 5′ extension. Nested PCR products were transcribed in vitro (T7 MEGA script kit; Ambion, Austin, TX) and the resulting double-stranded RNAs were annealed and processed with 30 units per reaction of human recombinant Dicer (Invitrogen) for 15 hr at 37°C. The 21mer siRNAs were separated from incompletely digested fragments using a succession of isopropanol precipitations and filtration on glass fiber plates (Nunc, Rochester, NY).

## Filopodia dynamics analysis

Neurons were imaged for 10 min every 60 s using a 63× objective. Changes in filopodia dynamics were assessed manually using ImageJ. In detail, dynamic protrusions (i.e., nodes and dynamic filopodia) were counted and normalized to filopodia that remained over the course of the acquisition (i.e., static filopodia). Dynamics was visualized using the Temporal Color-Code designed by Kota Miura (http://fiji.sc/wiki/index.php/Temporal-Color_Code).

## Ratiometric images of fluorescence intensity in dendritic nodes and actin patches

The software used for ratio-metric imaging has been previously described (*Tsai and Meyer, 2012*). In brief, a low-pass Gaussian filter was first applied to all images to suppress the noise while retaining the details of the fluorescent signals. Background subtraction was subsequently performed by (the value of each pixel)—(the mean value of the background within 40 μm of that pixel). To determine relative intracellular ArhGAP44 levels, ratio images were created dividing ArhGAP44 fluorescence over the cytosolic fluorescence.

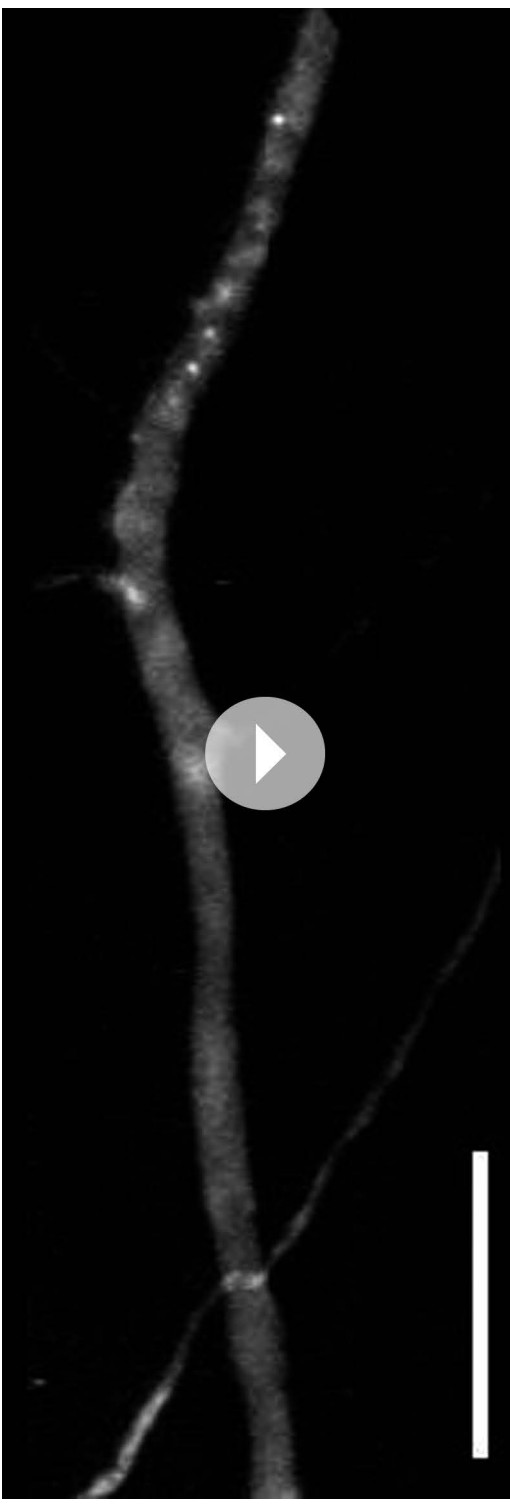

**Video 15**. Enrichment of the N-BAR domain of ArhGAP44 at the dendritic section that is in contact with nanocones. 3D rotation of neuron plated on nanocone-coated glass slide and transfected with the N-BAR domain of ArhGAP44. Note the punctate enrichment at the top where the dendrite is touching the nanocone surface. Scale bar is 10 μm.

## Analysis of relative protein concentration in dendritic patches

To measure the relative protein concentration at patches, average fluorescent intensities of the protein of interest (POI) and the cytosolic reference were measured in the patch and in the adjacent dendritic stretch. The background (BG) was determined separately for both channels using the average of four sectors outside the cell adjacent to the region of interest. The relative intensity of POI's was determined as $[(POI_{Patch} - POI_{BG})/(Cytosol_{Patch} - Cytosol_{BG})]/[(POI_{Dendrite} - POI_{BG})/(Cytosol_{Dendrite} - Cytosol_{BG})]$.

## Correlative IF/SEM image analysis

A micro-pattern was generated on glass slides and coated over night with PLL (0.1 mg/ml). Neurons were plated for 11 days and then fixed as described above for the FESEM analysis. Using the micro-pattern as reference points, individual neurons were then labeled with fluorescently tagged phalloidin and imaged on a 63× objective with a 1.5 Optovar. Cells were then sputter-coated with 70A of Au/Pd using a Denton Desk 11 Sputter Coater. Using the backscatter detector, individual micro-patterns were identified and used to navigate and identify individual previously imaged neurons as shown in *Figure 4—figure supplement 2*. SEM Images were then taken at 10,000× magnification and aligned with the immunofluorescent images. Finally, individual nodes were identified using the SEM images, and the average fluorescent intensity of phalloidin was measured in nodes and the adjacent dendritic stretches.

## Constructs and drugs

Full-length and the N-BAR domain constructs of ArhGAP44, F-tractin, NMHC-2B (Addgene Plasmid No. 11348), Rac1 and Cdc42 were previously described (*Wei and Adelstein, 2000*; *Heo and Meyer, 2003*; *Johnson and Schell, 2009*; *Galic et al., 2012*). The point mutation in ArhGAP44 (R291M) was introduced using the site-specific mutagenesis kit (200518; Stratagene, Cedar Creek, TX). All constructs were sequenced prior to use. ML-7 (sc-200557; Santa Cruz Biotechnology) was used at 10 μM (*Figure 5A*) and 50 μM (*Figure 6—figure supplement 1B*). Latrunculin A (428026; Cal Biochem, EMD Millipore, Billerica, MA) was used at 4 μM. Cytochalasin D (PHZ1063; Invitrogen) was used at 5 μM.

## Dug-induced changes in ArhGAP44 intensity

For individual neurons, z-stacks were acquired before and 20–30 min after addition of drugs.

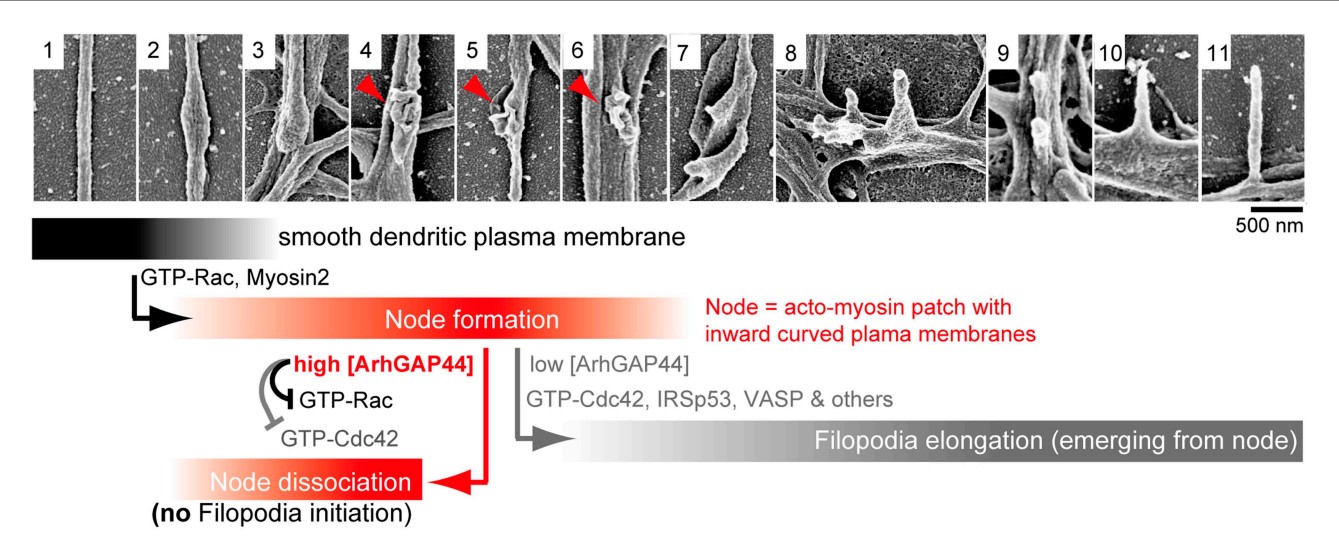

**Figure 6**. Proposed model of ArhGAP44-dependent regulation of exploratory dendritic filopodia initiation. Scanning electron micrographs of dendritic protrusions with and without filopodial protrusions aligned in a hypothetical time-line. Proposed model that myosin-dependent contractions in actin patches triggers local plasma membrane indentations (i.e., node formation) with curved membranes to which ArhGAP44 is recruited (frame 4–6, red). Increased local ArhGAP44 concentration limits Rac1-dependent actin polymerization and weakens or dissociates the node. Low ArhGAP44 concentration at nodes allows filopodia initiation that relies on Cdc42 and other factors (frame 7–11, gray). Scale bar, 500 nm.

The following figure supplement is available for figure 6:

**Figure supplement 1**. Control experiments supporting the proposed model.

For the analysis, a maximal projection was made and individual actin patches (= sites along the dendrite where the actin/cytosol ratio was >200% above the average ratio) were identified. Using a mask, the ArhGAP44/cytosol intensity within the actin patches was determined and normalized to ArhGAP44/cytosol ratio in the dendritic shaft.

### FKBP-Tiam1 assay

Dynamic translocation of Tiam1 with the FRB-FKB system in non-neuronal cells has been previously described (*Inoue et al., 2005*). Here, we cultured hippocampal neurons were quadruple-transfected with Lyn-FRB, CFP-FKBP-Tiam1 the YFP-tagged N-BAR domain of ArhGAP44 and the cytosolic reference mCherry using Lipofectamine 2000 (according to manufacturer's protocol). 24 hr later, individual cells were imaged before and after addition of 100 nM rapamycin (B0560; Sigma–Aldrich) with a 63× objective and a 1.5× Optovar module.

### Nanocone assay

Nanocone production has been previously described (*Jeong et al., 2011*; *Galic et al., 2012*). In brief, a 35–50 nm thin film of tin was deposited by heat evaporation on a glass coverslip at room temperature. The glass with the deposited tin was then exposed to a nitrogen gas environment with a low concentration of oxygen (about 1 part per million) at 350°C for 90 min. The annealing to the glass and the formation of the replicate nanocone shapes occurred during this heating step. In order to make the nanocone structures transparent, the glass coverslip with nanocones was further heated to a temperature 400°C for 3 hr in air. Labtek chambers were then mounted with nanocoated glass slides as previously described (*Jeong and Galic, 2014*), and neurons were subsequently cultured, transfected on DIV10, and imaged alive 24 hr later. 3D rotation (*Videos 14,15*) of confocal stacks was done in ImageJ.

### Atomic force microscopy

AFM images of nanocones done in tapping mode using commercial cantilevers on a JPK Nanowizard II instrument. Height analysis was performed using JPK image processing software.

## Statistics

p-values in all figures depict pair-wise comparisons and were evaluated using the Student's *t* test, with two tails and two-sample unequal variance. Error bars in all images represent SEM of the mean value. **p < 0.01.

## Acknowledgements

The authors thank Dr S Jeong for help with the fabrication of the glass-pattern, Dr L M Joubert from the Stanford Cell Sciences Imaging Facility for help with electron-microscopy, P Seleschik and A Ricker for help with atomic force microscopy, members of the Klämbt and Püschel labs for technical assistance, and members of the Meyer lab for helpful comments and critical discussions.

## Additional information

### Funding

| Funder | Grant reference number | Author |
|---|---|---|
| Deutsche Forschungsgemeinschaft | EXC 1003 | Milos Galic |
| National Institutes of Health | MH064801 | Tobias Meyer |
| National Institutes of Health | MH095087 | Tobias Meyer |
| National Institutes of Health | GM063702 | Tobias Meyer |

The funders had no role in study design, data collection and interpretation, or the decision to submit the work for publication.

### Author contributions

MG, Conception and design, Acquisition of data, Analysis and interpretation of data, Drafting or revising the article; F-CT, SB, Developed essential image analysis tools, Analysis and interpretation of data, Drafting or revising the article; SRC, Developed essential computational analysis tools, Analysis and interpretation of data, Drafting or revising the article; MM, Acquisition of data, Analysis and interpretation of data, Drafting or revising the article; TM, Conception and design, Analysis and interpretation of data, Drafting or revising the article

### Ethics

Animal experimentation: This study was performed in strict accordance with the recommendations in the Guide for the Care and Use of Laboratory Animals of the National Institutes of Health. All of the animals were handled according to approved institutional animal care and use committee (IACUC) protocols (#9998) at Stanford University.

## Additional files

### Supplementary file

• Supplementary file 1. Names and relative expression levels of 286 putative actin regulators. List depicting the expression levels of genes identified in the NCBI search in various tissues of the Human U133A/GNF1H Gene Atlas data set (gnf1h-gcrma unaveraged). Note that values are log-transformed.

### Major dataset

The following previously published dataset was used:

| Author(s) | Year | Dataset title | Dataset ID and/or URL | Database, license, and accessibility information |
|---|---|---|---|---|
| Su AI, Wiltshire T, Batalov S, Lapp H, Ching KA, Block D, et al. | 2004 | A gene atlas of the mouse and human protein-encoding transcriptomes | www.biogps.org | Freely accessible at www.biogps.org/. |

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
