## [Decision Letter]

[Editors’ note: this article was originally rejected after discussions between
the reviewers, but the manuscript was accepted after revisions and re-review.]

Thank you for choosing to send your work entitled “Dynamic Recruitment of Nadrin2
to Nanoscale Membrane Deformations Limits Exploratory Filopodia Initiation in
Neurons” for consideration at *eLife*. Your full submission has
been evaluated by Richard Losick (Senior editor), a Reviewing editor, and 2 peer
reviewers, and the decision was reached after discussions between the reviewers.

After careful examination of the results and multiple consultations, the reviewers and
the Reviewing editor decided that the results did not reach the level of significance
claimed by the authors and despite tackling an interesting problem, the main
interpretations would require a lot more work to meet our publication standard. The
reviewers’ comments follow for your consideration.

*Reviewer #1*:

The authors analyze here the Rac1 GAP and BAR-domain containing protein Nadrin2, which
they identified as an actin regulatory protein enriched in brain vs spinal cord and that
they show to be enriched in cortex. Overexpression and knock-down studies demonstrate
that this protein is a negative regulator of dendritic filopodia number. The GAP
activity contributes to this function, but is not required. Analyzing filopodia
dynamics, live imaging data upon knock-down and in overexpressing neurons support that
this protein causes newly formed filopodia to be unstable. Nadrin2 localizes to actin
rich dendritic patches (which are nicely analyzed using correlative scanning
EM/immunostaining), and imaging of the less active Nadrin2 GAP mutant supports that
filopodia can emerge from these dendritic Nadrin2 patches. Interestingly, Nadrin2
appears to be recruited dynamically and in a myosin light chain kinase-dependent manner
to these actin patches, which also contain Rac1. The model that emerges from these
results is that myosin-actin interactions at dendritic membranes cause membrane
indentations to form, which recruit Nadrin2 via its BAR domain to dynamically regulate
Rac1 and restrict filopodia formation. This provides evidence that stable filopodia
require not only positive signals to form but also the removal of negative regulators
such as Nadrin2, which is an interesting concept. Several key experimental questions
need to be addressed, however, to further strengthen this study.

1) How frequently do filopodia emerge from Nadrin2-positive patches? Are filopodia more
likely to emerge from these patches than from Nadrin-2 negative dendritic shaft areas?
This information will allow to assess how important a player Nadrin2 is in the control
of filopodia formation. If most emerging filopodia do not contain Nadrin2, this protein
is only relevant to control a subset of these protrusions.

2) Does overexpression or loss of Nadrin2 impact the number of dendritic spines? This is
important to assess the biological role of Nadrin2-mediated filopodia formation.

3) Can the authors show more directly that Nadrin2 is recruited to membranes in a
curvature-dependent manner?

4) Figure 2 only shows that Rac1 overexpression
promotes the formation of transient filopodia independent of Nadrin2. The statement that
“Nadrin2 limits Rac-dependent de novo filopodia formation” is not
supported by these data.

5) Knock-down neurons should be stained for Nadrin2 to confirm antibody specificity in
immunostaining.

6) The tissue distribution blot in Figure1 needs to be repeated for better signals.

Reviewer #2:

The paper identifies Nadrin2 as brain-enriched regulator of filopodia outgrowth in
neuronal dendrites, as overexpression of the wt strongly suppresses filopodial number.
More refined manipulations of Nadrin2 have a number of more moderate effects: it shows a
mild (50%) enrichment in actin patches along the dendrite compared to GFP, and its
knockdown prompts an increase in the number of quickly collapsing protrusions and a 20%
increase in steady state filopodia density. Its recruitment to patches is shown to
depend on its N-BAR domain, suggesting it is recruited to the membrane curvature
observed in detail by SEM in these patches. Pharmacological inhibition of Myosin II or
application of LatA or CytoD acutely reduced Nadrin2 at the patches, suggesting that the
status of actin and myosin in the patches is also involved in Nadrin2 targeting. The
model is that Nadrin2 recruitment suppresses Rac1 activity important for filopodial
extension, and this model is supported by weaker phenotypes of a Nadrin2 mutant that
retains only weak Rac1 GAP activity.

The paper presents a new model for suppression of protrusion formation, and postulates a
role for the membrane curvature seen in the filopodial birthplaces. These observations
highlight previously overlooked aspects of neuronal process formation. The combination
of live-cell imaging and correlative SEM is potentially quite powerful. However, the
live-cell approaches are not used to test the Nadrin2 model very directly, and though
the phenotype is compelling, overall I find the explanation offered by the model only
weakly supported by the available data.

My major concerns are as follows:

1) The authors overuse the terms “transient” and “dynamic”
to describe the Nadrin2 enrichment at patches. For example, “A major finding of
our study is that transient recruitment of Nadrin2 to actin patches...”…
“deletion of Nadrin2 curvature-sensitivity (Figure 3) as well as limiting action-myosin dependent contractile forces
(Figure 5) both prevented dynamic Nadrin2
recruitment.” “both treatments led to a significant reduction of Nadrin2
recruitment”

In fact, the time course of recruitment is never analyzed in the paper. There is only
one Figure 3 that shows any transience. No
treatment was analyzed in a manner that would have assessed the dynamic nature of the
recruitment. Perhaps the decreased but remaining colocalization is just as transient but
weak, or more short-lived but just as pronounced. Regarding myosin, the ML7 results
(Figure 5) seem to suggest only that
retention, not recruitment, of Nadrin2 are mediated by myosin II. It seems that
examining the time course of accumulating Nadrin2 with respect to the time of filopodial
extension would be required to evaluate the model. A more specific test would be to
compare the time course of active Rac1 in the patch with Nadrin2 levels. Similarly, the
curvature aspect could be addressed in detail specifically by timelapsing the N-BAR
construct during extension.

2) The authors state “the transient localization of Nadrin2 to patches but not to
extended filopodia argues that Nadrin2 limits initiation rather than elongation of newly
formed filopodia.”

Why doesn't this argue equally well that recruitment of Nadrin2 initiates
outgrowth? Loss of Nadrin in an elongating filopodium would seem to indicate that it
might in fact suppress elongation, not initiation. With the few examples available for
viewing, initiation seems to occur before loss of Nadrin. Clearly, an additional step of
mechanism would be required to explain the overexpression and knockdown phenotypes, but
I think it is too facile to say the live imaging supports this model.

3) The authors state “Our experiments identified the small GTPase Rac as the
principal target of Nadrin2 within actin patches (Figure 2—figure supplement 1).”

The indicated figure does not demonstrate this at all. The assumption is that Nadrin2
acts through Rac1 only, not Cdc42, but the alternative (interaction with Cdc42) is not
tested. Can this really be dismissed out of hand? Figure 1 tests the relative levels of GTP-Rac1, but does not test whether Nadrin2 or
its mutant alter levels of other potential targets (e.g. Cdc42).

A related but unstated presumption is that the dynamic assays of Figure 2 predict total protrusion number, and expressing Cdc42
would provide a test of that as effect of its overexpression is opposite to Rac1. Does
Cdc42 overexpression alter the dynamic measures in a way opposite to Rac1? Does it
decrease the number of nodes?

Presumably, dynamics were not measured following Nadrin2(wt) overexpression because
there are so few filopodia left, but shouldn't the Rac1+Nadrin2(wt) case be
examined, since Nadrin2(wt) should suppress the Rac1 phenotype? That is the claim in the
model of Figure 5, left column.

4) Surprisingly, the movies do not make a great case for this paper, and in fact in my
mind raise more questions than they answer. Technically, the intensity levels fluctuate
substantially, making me wonder whether small or thin structures such as nascent
filopodia or nodes could really be identified with confidence. One troubling example is
in the upper right panel, going from frame 1 to 2. The structure seemingly of most
interest is the node or protrusion in the middle of the process (under
“n2”), which appears to pop into existence. But the entire image brightens
in this panel, making it unclear whether to trust the relative intensity of the node.
Related to that type of phenomenon, some of the fluctuation is apparently due to an
inconsistent z span; e.g. in Movie 1, the overexpression of Nadrin2 panel clearly shows
the middle one of the processes fade out and then return, as it was insufficiently
spanned by the planes of the z sectioning. In the upper right panel, that particular
node could just have been captured more completely in bounds of the z stack rather than
appear or brighten in reality.

In terms of observations, it doesn't seem as though the control example in movie 1
demonstrates any protrusion formation at all. Am I missing something? Based on the
quantification in Figure 2, there should
be just as many forming and collapsing processes as there are processes that persist
through the movie, but I don't see this at all. At the very least, I'd say
that readers will need some help: formation, collapse, and unclassified changes should
probably be pointed out with markers of some sort.

Raw data (movies) of the Rac1 and Rac1+Nadrin2(R291M) experiments would be good to
see.

More raw data (stills or movies) from the key experiments in Figure 3 is needed.

5) In Figure 3, the accumulation of a set of SEM
images into a numbered series of panels portrayed as “stages” is
unnecessary and misleading-there is no evidence presented here or cited that the stages
correspond to a growth sequence. There is a near-total lack of information about how
these images were assessed or quantified, and what constituted a node or a protrusion or
any other feature.

---

## [Author Response]

We have substantially expanded description and analysis of filopodia initiation from
dendritic nodes, showing now in detail morphological rearrangements and kinetic behavior
of ArhGAP44 and actin at such sites. We also included in the revised version of the
manuscript experiments to investigate a possible regulation of Cdc42 by ArhGAP44 during
exploratory dendritic filpodia initiation. These experiments further validate the
proposed model that ArhGAP44 targets Rac1 in actin-patches that precede exploratory
filopodia formation. To directly demonstrate that inward membrane deformation is
sufficient for ArhGAP44 recruitment, we artificially indented the plasma membrane in
living neurons using cone-shaped nanostructures (nanocones) as well as a chemical
inducer of actin dynamics (a dimerization system we developed to rapidly pull a Rac GEF
from the cyctosol to the plasma membrane). Finally, we also included experiments to test
for a possible function of ArhGAP44 at other actin-rich neuronal structures, such as
dendritic spines and growth cones.

Together, the added studies further strengthened the validity of our major finding that
ArhGAP44 acts in a localized negative feedback that allows neurons to tune the frequency
with which new exploratory filopodia are initiated.

Comment on changed protein name: in compliance with the official protein nomenclature,
we are using in the revised version of the manuscript the protein symbol ArhGAP44
instead of Nadrin2.

Reviewer #1: *[…] Several key experimental questions need to be
addressed, however, to further strengthen this study*.

*1) How frequently do filopodia emerge from Nadrin2-positive patches? Are
filopodia more likely to emerge from these patches than from Nadrin-2 negative
dendritic shaft areas? This information will allow to assess how important a player
Nadrin2 is in the control of filopodia formation. If most emerging filopodia do not
contain Nadrin2, this protein is only relevant to control a subset of these
protrusions*.

In the revised version we now investigate the origins of dendritic protrusions in more
detail. The following three experiments were added:

We show that 83% ± 7% of all protrusions emerge from dendritic nodes (Figure 2—figure supplement 4).

We show that 89% ± 6% of all protrusions emerge from actin-rich patches (Figure 4—figure supplement 1). This
further strengthens the argument that dendritic nodes visible by light microscopy and
electron microscopy are local actin patches in dendrites.

100% of dendritic actin patches show enrichment for ArhGAP44. The average enrichment of
ArhGAP44 in dendritic actin-patches is 80% ± 8% over a cytosolic reference (Figure 4—figure supplement 3).

Together, these added experiments argue that the majority of dendritic protrusions are
emerging from dendritic nodes that have convoluted membrane invaginations and that show
a relative increase for both, actin and ArhGAP44 concentration.

*2) Does overexpression or loss of Nadrin2 impact the number of dendritic spines?
This is important to assess the biological role of Nadrin2-mediated filopodia
formation*.

To further explore the function of ArhGAP44 in aged neurons, we added the following two
experiments to the revised manuscript:

Full-length and the isolated N-BAR domain of ArhGAP44 both enrich in dendritic spines in
neurons (Figure 5—figure supplement 6).

Knockdown of ArhGAP44 in aged neurons (DIV17) increased the fraction of dynamic
filopodia-shaped vs. spine-shaped dendritic protrusions (Figure 5—figure supplement 7).

We did not include this data in the initial submission, as we felt that it does not
strengthen the core findings of this manuscript that is focused on the initiation of
exploratory filopodia from dendrites.

*3) Can the authors show more directly that Nadrin2 is recruited to membranes in
a curvature-dependent manner*?

Yes, we can. We have previously established an assay that relies on cone-shaped
nano-scale structures (nanocones) to deform the plasma membrane (PM) in live cells
(4): When cultured on
nanocones, adherent cell transiently deform the basal PM. This creates local sites with
positively curved PMs of up to 50nm diameter that allows investigating the functional
consequences of nano-scale membrane deformation under physiological conditions in live
cells. In the revised version we added the following experiments:

We show that the isolated N-BAR domain of ArhGAP44 forms puncta selectively above
nanocone-induced membrane-deformation in the basal PM of primary hippocampal neurons
(Figure 5—figure supplement 4 and
Videos 14 and 15).

We show that enrichment of ArhGAP44 over nanocone-induced PM-deformations does not
correlate with enrichment of the PM (Figure 5—figure supplement 3) or of actin (Figure 5—figure supplement 3).

Together, these experiments argue that nanocones locally deform the PM and that the
resulting local high PM-curvature is sufficient for ArhGAP44 enrichment in neurons.

*4)*
Figure 2
*only shows that Rac1 overexpression promotes the formation of transient
filopodia independent of Nadrin2. The statement that “Nadrin2 limits
Rac-dependent de novo filopodia formation” is not supported by these
data*.

We thank the referee for pointing out that the title of the figure legend is misleading.
We have rephrased the figure title to: ´Knockdown of ArhGAP44 and overexpression of
Rac1 both increase de novo filopodia formation´.

*5) Knock-down neurons should be stained for Nadrin2 to confirm antibody
specificity in immunostaining*.

The following two experiments were added to confirm the antibody specificity:

Overexpression control: Neurons transfected with ArhGAP44(R291M) for 24h, fixed and
stained with an antibody directed against ArhGAP44 show a >8-fold increase in
fluorescence intensity compared to non-transfected cells (Figure 3—figure supplement 1).

Knockdown control: When stained with an antibody directed against ArhGAP44, neurons
transfected with siRNA directed against ArhGAP44 show reduced fluorescence intensity
compared to cells transfected with a control siRNA (Figure 3—figure supplement 1).

*6) The tissue distribution blot in Figure1 needs to be repeated for better
signals*.

The blot shown in Figure 1 has been
replaced.

Reviewer #2: *My major concerns are as follows*:

*1) The authors overuse the terms “transient” and
“dynamic” to describe the Nadrin2 enrichment at patches. For example,
“A major finding of our study is that transient recruitment of Nadrin2 to
actin patches...”…“deletion of Nadrin2 curvature-sensitivity
(*Figure 3*)
as well as limiting action-myosin dependent contractile forces (*Figure 5*) both prevented
dynamic Nadrin2 recruitment.”… “both treatments led to a
significant reduction of Nadrin2 recruitment*”

*In fact, the time course of recruitment is never analyzed in the paper. There is
only one*
Figure 3
*that shows any transience. No treatment was analyzed in a manner that would have
assessed the dynamic nature of the recruitment. Perhaps the decreased but remaining
colocalization is just as transient but weak, or more short-lived but just as
pronounced. Regarding myosin, the ML7 results (*Figure 5*) seem to suggest only that
retention, not recruitment, of Nadrin2 are mediated by myosin II. It seems that
examining the time course of accumulating Nadrin2 with respect to the time of
filopodial extension would be required to evaluate the model. A more specific test
would be to compare the time course of active Rac1 in the patch with Nadrin2 levels.
Similarly, the curvature aspect could be addressed in detail specifically by
timelapsing the N-BAR construct during extension*.

(A) Overuse of the terms ‘transient’ and ‘dynamic’. We have
substantially reduced the use of both terms in the text, using them now only 2 times in
the whole manuscript to describe ArhGAP44.

(B) Curvature-aspect of protein recruitment. In the revised version of the manuscript,
we added a series of experiments to investigate the curvature-dependence of ArhGAP44
enrichment (see also Referee #1, point 3):

We show that the isolated N-BAR domain of ArhGAP44 forms puncta selectively above
nanocone-induced membrane-deformation in the basal PM of primary hippocampal neurons
(Figure 5—figure supplement 4 and
Videos 14 and 15).

We show that enrichment of ArhGAP44 over nanocone-induced PM-deformations does not
correlate with enrichment of the plasma membrane (Figure 5—figure supplement 3) or of actin (Figure 5—figure supplement 3).

These experiments provide evidence that inward plasma membrane deformation is sufficient
for enrichment of ArhGAP44 to inward deformed plasma membranes in neurons.

(C) Time course of ArhGAP44 enrichment. The following four experiments were added:
ArhGAP44 is recruited to contracting structures: We added a synthetic approach that we
developed earlier, where we used a small molecule to rapidly recruit a Rac GEF to the PM
and activate Rac in order to increase actin polymerization (Figure 5—figure supplement 2 and Video 13). In this approach, we observe enrichment of the
isolated N-BAR domain of ArhGAP44 at retracting actin-rich structures induced along the
dendritic shaft. Together with the experiment where we inhibit MLCK using ML-7 (Figure 5) and show that recruitment of ArhGAP44
requires myosin contraction, this argues that the increase in local ArhGAP44
concentration is caused by increased formation of acto-myosin-dependent inward membrane
deformation.

Kinetic analysis of actin and ArhGAP44 during filopodia initiation. The following
experiments were added: we find that 83% ± 7% of all protrusion emerge from
dendritic nodes (Figure 2—figure supplement 3), that 89% ± 6% of all protrusions emerge from actin patches (Figure 4—figure supplement 1), and that
ArhGAP44 is enriched in 100% of actin patches (Figure 4—figure supplement 3). These experiments argue that the majority of
dendritic protrusions are emerging from dendritic nodes that are enriched both in actin
and in ArhGAP44.

Together, these added experiments further strengthen the statement that ArhGAP44 is
recruited to contracting actin patches within nodes that precede filopodia
elongation.

(D) Retention vs. Recruitment. We propose that ArhGAP44 diffuses trough the cytosol by
Brownian movement, and that enrichment of ArhGAP44 at nodes is caused by binding of the
protein to inward membrane deformations that transiently form at such sites due to
myosin-dependent contraction of membrane-associated actin cables. Binding of N-BAR
domain proteins (such as ArhGAP44) has been shown to depend (i) on increased
electrostatic interactions between negatively charged lipid head groups in curved
membranes and positively charged amino acids of the banana-shaped protein dimer facing
the membrane, and (ii) the insertion of an amphipatic helix present in all N-BAR domain
proteins into membrane-imperfections that predominantly occur in curved lipid bilayers
(14; 66). Enrichment of N-BAR domain proteins at
curved membranes is thus believed to be due to an increase in binding affinity that may
result molecularly from a reduced off-rate and possibly an increased on-rate. As the
term ‘recruitment’ does not imply active directed transport (it is often
used to describe a diffusion mediated retention at local sites), we feel that it is
appropriate to use this term since it describes the observed dynamic increase in local
concentration of ArhGAP44 when new actin patches are formed.

*2) The authors state “the transient localization of Nadrin2 to patches
but not to extended filopodia argues that Nadrin2 limits initiation rather than
elongation of newly formed filopodia*.”

*Why doesn't this argue equally well that recruitment of Nadrin2 initiates
outgrowth? Loss of Nadrin in an elongating filopodium would seem to indicate that it
might in fact suppress elongation, not initiation. With the few examples available
for viewing, initiation seems to occur before loss of Nadrin. Clearly, an additional
step of mechanism would be required to explain the overexpression and knockdown
phenotypes, but I think it is too facile to say the live imaging supports this
model*.

We agree with the referee that it is a valid question whether ArhGAP44 functions more as
an inhibitor of filopodia initiation vs. an inhibitor of filopodia elongation/outgrowth:
We considered that an inhibitor of filopodia elongation/outgrowth would be expected to
localize together with positive regulators of actin dynamics (e.g. MENA or IRSp53) at
the tip of extending filopodia where actin-polymerization in growing filopodia occurs.
In contrast, an inhibitor of filopodia initiation would be expected to be present at the
´birthplace´ of filopodia, and to dissociate once elongation/outgrowth begins.
The localization of ArhGAP44 all across actin patches, which our data argues is caused
by acto-myosin dependent inward curved plasma membrane deformation, suggests that it
reduces Rac and Cdc42 activity globally across the patch which we think explains the
lower frequency of filopodia extension. The dissociation from extending filopodia (with
outward membrane deformation) can best be explained by the change in curvature (all
negative) in the extending membrane tubes. Based on the selectivity of N-BAR domain
proteins to positively curved membranes, we therefore think that our data is consistent
with a role of ArhGAP44 in regulating filopodia initiation. In the revised version of
the manuscript we discuss this now in the text.

As an added note: the possibility that ArhGAP44 may have additional functions in nodes
(e.g. inhibit filopodia initiation AND elongation/outgrowth) will be discussed in more
detail below in point 4.

*3) The authors state “Our experiments identified the small GTPase Rac as
the principal target of Nadrin2 within actin patches (*Figure 2—figure supplement 1*).” The indicated figure does not demonstrate this
at all*.

We agree. The statement has been replaced.

*4) The assumption is that Nadrin2 acts through Rac1 only, not Cdc42, but the
alternative (interaction with Cdc42) is not tested. Can this really be dismissed out
of hand?*
Figure 1
*tests the relative levels of GTP-Rac1, but does not test whether Nadrin2 or its
mutant alter levels of other potential targets (e.g. Cdc42)*.

*A related but unstated presumption is that the dynamic assays of*
Figure 2
*predict total protrusion number, and expressing Cdc42 would provide a test of
that as effect of its overexpression is opposite to Rac1. Does Cdc42 overexpression
alter the dynamic measures in a way opposite to Rac1? Does it decrease the number of
nodes*?

This is an excellent question. Considering the dual specificity of ArhGAP44 to Rac1 and
Cdc42 (as we cite in the paper), ArhGAP44 also likely has an added function in
regulating Cdc42 during filopodia initiation (it is well established that Cdc42 is
critically involved in filopodia formation (40; 42; 36)). In the revised version, we
have added 3 experiments as well as raw material to characterize the potential interplay
between ArhGAP44 and Cdc42:

ArhGAP44 can hydrolyze GTP-Cdc42: Consistent with previous reports, we find hydrolysis
of GFP-Cdc42 by ArhGAP44 (Figure 2—figure supplement 5).

Overexpression of Rac1(wt) but not of Cdc42(wt) phenocopies protrusion dynamics upon
knockdown of ArhGAP44: We have previously shown that knockdown of ArhGAP44 increases
density (Figure 1, yellow) and dynamics (Figure 2, yellow) of dendritic protrusions, while
overexpression of ArhGAP44 reduces protrusion density (Figure 1, blue) and protrusion dynamics (Figure 2, blue). As ArhGAP44 is a RhoGAP, overexpression of the small GTPase
that is targeted by ArhGAP44 should phenocopy the knockdown of ArhGAP44. We previously
showed that overexpression of Rac1(wt) increases density (Figure 2—figure supplement 5, red) and dynamics (Figure 2, red) of dendritic protrusions, and
rescues reduced protrusion dynamics upon overexpression of ArhGAP44 (Figure 2, purple). For Cdc42, we showed in the
initial submission that overexpression of Cdc42(wt) did not increase protrusion density
(Figure 2—figure supplement 5,
green). We have now added experiments that show that overexpression of Cdc42(wt) has no
effect on protrusion dynamics, and does not rescues reduced protrusion dynamics upon
overexpression of ArhGAP44(R291M) (Figure 2—figure supplement 5). The latter suggests that Rac regulation is
more important but does not exclude an additional regulatory role of Cdc42.

Overexpression of Rac1(wt) but not of Cdc42(wt) phenocopies protrusion morphology upon
knockdown of ArhGAP44: In the revised version of the manuscript, we added raw material
showing that overexpression of Rac1 but not of Cdc42 phenocopied the ArhGAP44 knockdown
causing the formation of dynamic dendritic nodes and filopodia (Figure 2—figure supplement 5 and Videos 3 and 4, 6–10). Again, this suggest for a more
important role of Rac but does not exclude an added regulation of ArhGAP44 on Cdc42.

Together, while these results argue for ArhGAP44-dependent regulation of Rac1 in actin
patches, we agree with the referee that a discussion about the function of Cdc42 has
been missing in the initial manuscript. Following discussion on the role of Cdc42 have
been added to the text: Cdc42 acts as an activator of Irsp53 (37), promoting IRSp53-dependent enrichment and
clustering of VASP and other factors to drive actin assembly in elongating filopodia
(35). Consistently,
knockdown of Cdc42 substantially reduces filopodia formation in neurons (52). Intriguingly,
overexpression of Cdc42 is not sufficient to initiate filopodia formation in neurons
(Figure 2—figure supplement 5, see
also (53)) or in other cell
lines (37). This has led to
the hypothesis that elongation of filopodia is a combinatorial process requiring
multiple factors (37). We
propose that signal integration at actin patches controls this decision of filopodia
elongation. Considering that actin-patch formation occurs before filopodia elongation,
this argues for a 2-step process where Rac1-induced patch formation (and
ArhGAP44-dependent regulation thereof) precedes Cdc42-induced filopodia elongation
(Figure 6). However, since ArhGAP44 shows dual
specificity for Rac1 and Cdc42, both steps will be limited by recruitment by ArhGAP44 to
actin patches.

*5) Presumably, dynamics were not measured following Nadrin2(wt) overexpression
because there are so few filopodia left, but shouldn't the Rac1+Nadrin2(wt)
case be examined, since Nadrin2(wt) should suppress the Rac1 phenotype? That is the
claim in the model of*
Figure 5*, left
column*.

Expression of ArhGAP44(wt) rapidly triggers varicosity formation and cell death (Figure 1—figure supplement 6), likely due
to excessive levels of the enzyme. Consequentially, either enzymatic efficiency or
expression levels need to be reduced to study ArhGAP44 function in neurons. Since
western blot analysis and live cell experiments show that ArhGAP44(R291M) is less potent
than ArhGAP44(wt) but still active (Figure 1 and
Figure 1—figure supplement 6), we
decided to use ArhGAP44(R291M) for the synthetic rescue experiments. In contrast to a
system that would rely on reduced expression of ArhGAP44(wt), this approach has the
additional advantage of providing a strong fluorescence signal that is critical to
identify transfected cells and study sub-cellular protein localization.

*6) Surprisingly, the movies do not make a great case for this paper, and in fact
in my mind raise more questions than they answer. Technically, the intensity levels
fluctuate substantially, making me wonder whether small or thin structures such as
nascent filopodia or nodes could really be identified with confidence. One troubling
example is in the upper right panel, going from frame 1 to 2. The structure seemingly
of most interest is the node or protrusion in the middle of the process (under
“n2”), which appears to pop into existence. But the entire image
brightens in this panel, making it unclear whether to trust the relative intensity of
the node. Related to that type of phenomenon, some of the fluctuation is apparently
due to an inconsistent z span; e.g. in Movie 1, the overexpression of Nadrin2 panel
clearly shows the middle one of the processes fade out and then return, as it was
insufficiently spanned by the planes of the z sectioning. In the upper right panel,
that particular node could just have been captured more completely in bounds of the z
stack rather than appear or brighten in reality*.

*In terms of observations, it doesn't seem as though the control example in
movie 1 demonstrates any protrusion formation at all. Am I missing something? Based
on the quantification in*
Figure 2*, there
should be just as many forming and collapsing processes as there are processes that
persist through the movie, but I don't see this at all. At the very least,
I'd say that readers will need some help: formation, collapse, and unclassified
changes should probably be pointed out with markers of some sort*.

*Raw data (movies) of the Rac1 and Rac1+Nadrin2(R291M) experiments would be
good to see*.

*More raw data (stills or movies) from the key experiments in*
Figure 3
*is needed*.

(A) Added raw material: We agree with the reviewer that addition of more raw data would
be helpful to (i) explain what types of dendritic protrusions exist, and (ii) show how
these protrusions are affected by overexpression and knockdown of ArhGAP44 or the small
GTPases Rac1 and Cdc42. To better explain and illustrate the main findings of the
analysis shown in Figure 2, the following 11 raw
data movies and 3 figure supplements have been added:

To illustrate different types of protrusions formed on dendrites, and better explain the
analysis used in Figure 2:

Timelapse examples of static and dynamic protrusion types characterized in Figure 2 (Figure 2—figure supplement 2).

Raw data movie showing stable dendritic filopodia (Video 1, filopodia lasts for >40 minutes).

Raw data showing formation of dynamic dendritic nodes (Figure 2—figure supplement 3 and Video 5).

To illustrate the findings of Figure 2:

Raw data movie showing increased dendritic node and reduced protrusion formation upon
ArhGAP44(R291M) overexpression (Video 2).

Raw data showing increased dendritic node and protrusion formation upon knockdown of
ArhGAP44 (Figure 2—figure supplement 5
and Videos 3 and 4).

Raw data movie showing dendritic protrusion emerging from dendritic nodes upon knockdown
of ArhGAP44 (Video 6).

To illustrate the findings of Figure 2:

Raw data examples showing abnormal dendritic node and filopodia formation upon Rac1
overexpression (Figure 2—figure supplement 5 and Videos 7 and 8).

Raw data examples showing abnormal dendritic filopodia formation upon Cdc42
overexpression (Figure 2—figure supplement 5 and Videos 9 and 10).

Raw data movie showing synthetic rescue of ArhGAP44(R291M)-dependent reduction in
filopodia formation by co-overexpression of Rac1(wt) (Video 11).

(B) Protrusions ‘pop into existence’? As the referee correctly state, we
only observe structures in the illuminated plane of the confocal microscope. This means
that dynamic dendritic sections that move in z-direction during the 10 minute long
acquisition period may cause filopodia to leave the confocal plane. To investigate the
frequency and origins of protrusions that ‘pop into existence, the following
experiments were added:

Timelapse analysis of filopodia formation using a shorter acquisition interval: As shown
in Figure 2—figure supplement 4, 57%
± 8% of all filopodia elongate from nodes, while the remaining filopodia appear to
emerge directly from the dendritic shaft (i.e. ‘pop into existence’) using
an acquisition interval of 60 seconds (between frames). However, when the acquisition
interval was reduced from 60 seconds to 15 seconds (i.e. between frames), the fraction
of filopodia emerging from nodes increased to 83% ± 7%.

Time series showing dendritic filopodia emerging from node (Figure 2—figure supplement 4). This example shows that
the acquisition interval of 60 seconds is sufficient to detect protrusions, but may in
some cases miss nodes (Figure 2—figure supplement 4, compare columns 2 and 4).

These added experiments provide evidence that the majority of filopodia that ‘pop
into existence’ are reflective of filopodia emerging from dendritic nodes that
were to short-lived to be captured using a 60-second interval. We consider the remaining
17% of filopodia that ‘pop into existence’ to be a mixure of (i) filopodia
emerging from nodes in less than 15 seconds (i.e. between frames), (ii) filopodia
emerging directly from the dendritic shaft, or, more likely, (iii) filopodia that appear
due to changes in the focal plane of the dendritic tree.

*7) In*
Figure 3*, the
accumulation of a set of SEM images into a numbered series of panels portrayed as
“stages” is unnecessary and misleading-there is no evidence presented
here or cited that the stages correspond to a growth sequence. There is a near-total
lack of information about how these images were assessed or quantified, and what
constituted a node or a protrusion or any other feature*.

The reviewer raised a valid point. As the sequence of electron micrographs that was
previous shown in Figure 3 is an interpretation
of data, it has now been integrated in the proposed model (Figure 6). To better describe how the images in Figure 1 were quantified, the following
supplemental figure panels and methods section have been added:

Figure 1—figure supplement 5 illustrates
how dendritic protrusions were quantified in scanning electron micrographs.

New chapter in the Material and methods section: Quantification of Protrusion Types
using Scanning Electron Micrographs. Neurons were cultured on glass slides for various
periods of time (3, 10 and 17 days), fixed and prepared for SEM as described above.
Using low resolution (1000x magnification), individual neurons were identified (Figure 1—figure supplement 5, left panel).
Starting from the soma, initial segments of the dendritic arbors were imaged at high
resolution (10’000x), and individual protrusions were classified based on
morphology (Figure 1—figure supplement 5, right panel). Only the proximal 50-60 μm of the dendritic arbors that
can clearly be associated to a particular neuron were analyzed. Examples of dendritic
nodes are shown in Figure 1—figure supplement 5.